# Voluntary and involuntary contributions to perceptually guided saccadic choices resolved with millisecond precision

**Emilio Salinas\*, Benjamin R Steinberg, Lauren A Sussman, Sophia M Fry, Christopher K Hauser, Denise D Anderson, Terrence R Stanford**

Department of Neurobiology and Anatomy, Wake Forest School of Medicine, Winston-Salem, United States

**Abstract** In the antisaccade task, which is considered a sensitive assay of cognitive function, a salient visual cue appears and the participant must look away from it. This requires sensory, motor-planning, and cognitive neural mechanisms, but what are their unique contributions to performance, and when exactly are they engaged? Here, by manipulating task urgency, we generate a psychophysical curve that tracks the evolution of the saccadic choice process with millisecond precision, and resolve the distinct contributions of reflexive (exogenous) and voluntary (endogenous) perceptual mechanisms to antisaccade performance over time. Both progress extremely rapidly, the former driving the eyes toward the cue early on (~100 ms after cue onset) and the latter directing them away from the cue ~40 ms later. The behavioral and modeling results provide a detailed, dynamical characterization of attentional and oculomotor capture that is not only qualitatively consistent across participants, but also indicative of their individual perceptual capacities.

DOI: https://doi.org/10.7554/eLife.46359.001

## Introduction

Neuroscience aims to explain macroscopic behavior based on the microscopic operation of distinct neural circuits, and this requires carefully designed tasks that expose the relationship between the two. In the case of the antisaccade task (*Coe and Munoz, 2017*; *Munoz and Everling, 2004*), in which participants are instructed to withhold responding to a visual cue in favor of programming a saccade to a diametrically opposed location, performance relies heavily on frontal cortical mechanisms associated with cognitive control (*Guitton et al., 1985*; *Everling and Fischer, 1998*; *Munoz and Everling, 2004*; *Condy et al., 2007*; *Luna et al., 2008*; *Hakvoort Schwerdtfeger et al., 2012*), and the paradigm is considered to be a sensitive assay of impulsivity and executive function in general. Indeed, the mean reaction time (RT) and overall error rate in the antisaccade task are frequently used as biomarkers for cognitive development (*Klein and Foerster, 2001*; *Luna et al., 2008*; *Coe and Munoz, 2017*) and, in clinical settings, for mental dysfunction (*Everling and Fischer, 1998*; *Munoz et al., 2003*; *Hutton and Ettinger, 2006*; *Antoniades et al., 2015*; *Wiecki et al., 2016*).

The antisaccade task pits against each other two fundamental processes, one involuntary and the other voluntary. On one hand, the sudden appearance of a salient visual stimulus automatically attracts spatial attention (*Theeuwes, 1991*; *Theeuwes et al., 1998*; *Ruz and Lupiáñez, 2002*; *Busse et al., 2008*; *Theeuwes, 2010*; *Carrasco, 2011*; *Aagten-Murphy and Bays, 2017*) to produce either a covert shift ('attentional capture') or an overt saccade ('oculomotor capture'). In either case, the effect is described as bottom-up or exogenous, and is thought to be fast and transient. On the other hand, programming a saccade away from a cue is a top-down or endogenous process that

**\*For correspondence:**
esalinas@wakehealth.edu

**Competing interest:** See
page 18

**Reviewing editor:** Daeyeol Lee,
Yale School of Medicine, United
States

**eLife digest** How do you decide what to do next? Your behavior at any given moment is usually the result of a competition between internal and external factors. Internal factors include your existing plans, goals and knowledge. External factors include events happening in the world around you. When out driving, for example, you check zebra crossings because you know that pedestrians could be present. But you look at stoplights because your eyes are drawn automatically to their changing colors.

Scientists can study this competition between internal and external factors using a simple laboratory task. A single spot of light appears in the dark, and your job is to look away from it. The instruction is simple and yet carrying it out requires willful effort. This is because your automatic response is to look at any stimulus that suddenly appears. Overcoming this automatic response requires similar thought processes to those that help someone resist eating that second piece of chocolate.

However, the competition between automatic and voluntary visual processes is over in a fraction of a second, which makes it difficult to analyze. Salinas et al. therefore modified the "look-away" task by asking participants to respond under time pressure. This tweak makes it possible to track – with millisecond precision – voluntary and automatic influences on performance. The results revealed that the eyes are automatically drawn to the cue about 100 milliseconds after it appears. The separate voluntary process that directs the eyes away from the cue arises about 40 milliseconds later.

Salinas et al. observed these voluntary and involuntary components in every healthy volunteer tested. But there were also differences between individuals in how effectively they could look away from the cue. This is important because the automatic draw of salient stimuli determines what you pay attention to, as well as what you look at. Future studies could use the modified version of the look-away task to examine whether this automatic pull of attention, and the ability to resist it, differs in individuals with disorders like ADHD.

DOI: https://doi.org/10.7554/eLife.46359.002

likely summons several mechanisms, such as working memory (*Roberts et al., 1994*; *Lavie and De Fockert, 2005*) and endogenous attention (*Godijn and Theeuwes, 2002*; *Theeuwes, 2010*; *Carrasco, 2011*), which are thought to be slower and to require a sustained cognitive effort. Thus, the rationale for the task is sound — the timing and intensity of the conflict between bottom-up and top-down mechanisms should correlate with behavior, and with the dynamics of the underlying attentional and oculomotor neural circuits.

There is a problem, however: such conflict must unfold very quickly. First, exogenous attention is thought to be mediated by visually-driven responses in oculomotor areas such as the frontal eye field (FEF) and superior colliculus (SC), which have latencies of at least 50 ms (*Gottlieb and Goldberg, 1999*; *Bisley et al., 2004*; *Thompson et al., 2005*; *Ipata et al., 2006*; *Joiner et al., 2017*; *White et al., 2017*; *Chen et al., 2018*). And second, spatial attention can be endogenously shifted roughly 150 ms after a relevant cue is provided (*Kim and Cave, 1999*; *Ogawa and Komatsu, 2004*; *Busse et al., 2008*; *Theeuwes, 2010*; *Markowitz et al., 2011*). This suggests that the competition between exogenous and endogenous responses evolves in less than 100 ms. The usual behavioral metrics of mean RT and overall accuracy are thus unlikely to yield a clean characterization of this competition, because they can be traded against each other and reflect the end results of numerous operations (perceptual, motor, cognitive) that contribute to a much longer choice process (indeed, below we show that such metrics are severely confounded). How can this problem be overcome?

The solution is to make the task urgent. The compelled antisaccade task requires subjects, humans in this case, to begin programming a saccade before knowing the direction of the correct response, and to use later arriving information about cue location to appropriately modify the ongoing motor plans. Urgency allows us to generate a special psychometric function, the 'tachometric curve,' which tracks success rate as a function of the perceptually relevant time interval, the raw processing time (rPT, measured between cue presentation and saccade onset). We find that, for the compelled antisaccade task, the tachometric curve takes on a unique shape: within a narrow rPT

range, the curve yields a pronounced dip to below-chance performance in which the exogenous capture by the cue is so strong that the success rate approaches 0%; thereafter, however, endogenous control takes over, and the fraction of correct saccades to the 'anti' location increases rapidly. The experimental data were comprehensively replicated by a neurophysiologically based model of the saccadic choice process, with the combined results providing a remarkably detailed account of how reflexive and voluntary mechanisms compete over time to determine task performance.

## Results

### Urgent antisaccade behavior is characterized by strong but temporally constrained oculomotor capture

Akin to an athlete anticipating the trajectory of a ball that must be caught or struck, the participant in the compelled antisaccade task must begin programming a movement in advance of the relevant sensory information, and must quickly interpret the later arriving visual cue to modify the developing motor plan(s) on the fly. In the sequence of task events (*Figure 1a*), the key step is the early offset of the fixation spot, which means 'respond now!' This go signal is given first, before the cue, which is revealed after an unpredictable gap period. The cue appears randomly to the left or right of fixation, and the participant is instructed to make an eye movement away from it, to the diametrically opposite location — but for this response to be correct, the saccade must be initiated within 450 ms of the go signal. Thus, due to urgency, that is, time pressure, the participant must begin planning (and may even execute) a motor response before knowing what the correct choice is. This design, in which motor and perceptual processes are meant to run concurrently, stands in contrast to the easy, non-urgent version of the task (*Figure 1b*), in which delivery of the cue before the go signal allows more time for the perceptual process to be completed before saccade onset.

As in other perceptually based urgent tasks (*Becker and Jürgens, 1979*; *Stanford et al., 2010*; *Salinas and Stanford, 2013*; *Scerra et al., 2019*), the cue viewing time or rPT (computed as RT − gap, or RT + delay; *Figure 1*) is the crucial variable here, because it specifies how much time is available for detecting and analyzing the cue in each trial. With little or no time to see the cue, the success rate cannot rise above chance, but as the viewing time increases, performance is expected to improve. Using multiple gap values (0–350 ms) ensures full coverage of the relevant rPT range. When the probability of making a correct choice is plotted as a function of rPT — a behavioral metric that we refer to as the tachometric curve — the result is a millisecond-by-millisecond readout of the

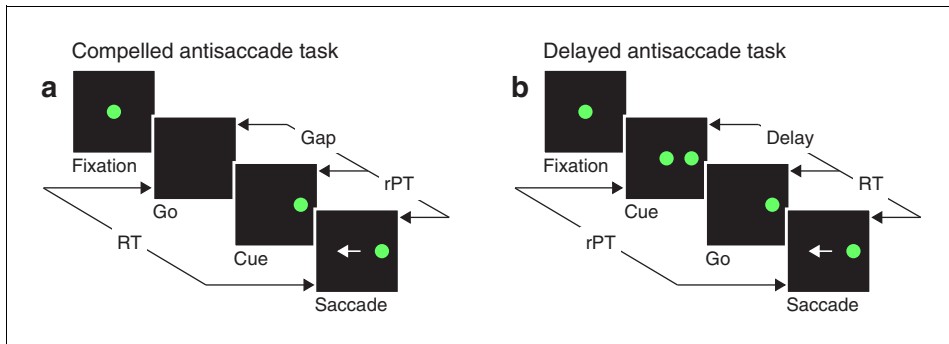

**Figure 1.** Urgent and non-urgent variants of the antisaccade task. (a) The compelled antisaccade task. After a fixation period (150, 250, or 350 ms), the central fixation point disappears (Go), instructing the participant to make an eye movement to the left or to the right (±10°) within 450 ms. The cue is revealed (Cue) after a time gap that varies unpredictably across trials (Gap, 0–350 ms). The correct response is an eye movement (Saccade, white arrow) away from the cue, to the diametrically opposite, or anti, location. (b) The delayed antisaccade task. In this case, the cue is shown before the go signal, during fixation. The interval between cue onset and fixation offset varies across trials (Delay, 100 or 200 ms). In all trials, the reaction time (RT) is measured between the onset of the go signal and the onset of the saccade, whereas the raw processing time (rPT) is measured between cue onset and saccade onset.

DOI: https://doi.org/10.7554/eLife.46359.003

evolving perceptual decision (*Becker and Jürgens, 1979*; *Stanford et al., 2010*; *Shankar et al., 2011*; *Salinas and Stanford, 2013*; *Seideman et al., 2018*).

For the compelled antisaccade task, the tachometric curve exhibits a unique, non-monotonic shape that reflects the interaction between early involuntary and later voluntary processes (*Figure 2*). For rPTs shorter than 90 ms, participants perform at chance, as expected. Shortly thereafter, the initial influence of the cue manifests as a pronounced drop in performance, as participants erroneously direct a large proportion of their saccades toward the cue. This dip, which we refer to as the 'vortex,' is short-lived (visible for rPTs of 100–140 ms approximately; *Figure 2*, gray shades), but it is so abrupt and occurs so reliably over a consistent range of rPTs, that it reaches nearly 0% correct even in the data pooled from all six participants (*Figure 2*, main panel). In trials in which the rPT falls inside this narrow interval, it is almost impossible to avoid looking at the cue.

As rPT increases beyond 140 ms, the success rate rises and gradually approaches an asymptote, as participants direct a progressively larger proportion of their saccades to the correct, anti location. This rise in performance is remarkable in that it is extremely fast: for the pooled data (*Figure 2*, main panel) the tachometric curve goes from 0.25 to 0.75 in 18 ms, and from 0.10 to 0.90 in only 37 ms. For some of the participants, the process is faster (*Figure 2*, P1, P2, P4). The asymptotic fraction of correct responses is close to 1 (the lowest across participants was 0.978), which indicates that the participants understood the instructions and could perform the task almost perfectly — given

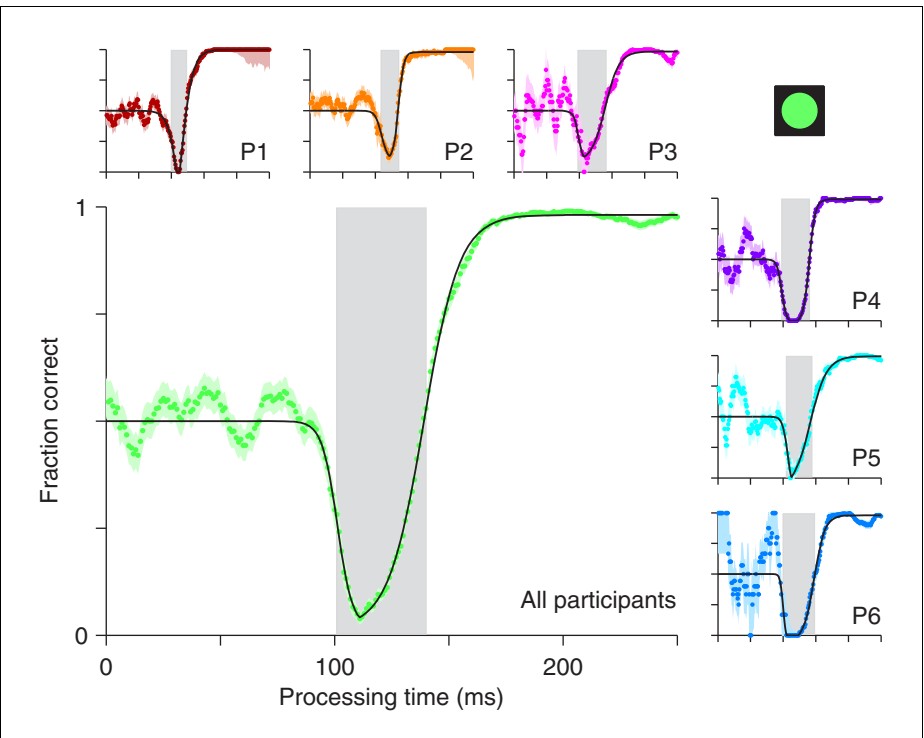

**Figure 2.** Perceptual performance in the compelled antisaccade task demonstrates a vortex. Each panel shows a tachometric curve, that is, a plot of the probability of making a correct response as a function of rPT, or cue-viewing time. Colored points are experimental results in overlapping time bins (bin width = 15 ms); light shades indicate ± 1 SE from binomial statistics; black lines are continuous, analytical functions fitted to the data. The vortex is the part of the curve for which saccades are highly likely to be captured and performance drops below chance. It is demarcated by gray shades for individual participants (small panels, P1 – P6) and for the aggregate data set (large panel, All participants). Results are from trials (between 1366 and 1534 per participant) in which the high-luminance cue (icon) was shown.

DOI: https://doi.org/10.7554/eLife.46359.004

The following figure supplement is available for figure 2:

**Figure supplement 1.** Performance in non-urgent antisaccade trials.
DOI: https://doi.org/10.7554/eLife.46359.005

enough time. Consistent with this, in easy, non-urgent antisaccade trials (*Figure 1b*) the fraction correct was also close to 1 (median = 0.992). However, because the processing times it generates are so long, the easy version of the task only provides a glimpse of the capture phenomenon, if anything (*Figure 2—figure supplement 1*).

## Antisaccade performance varies with cue luminance

The characteristic shape of the tachometric curve likely results from the interplay between a reflexive and a voluntary mechanism, both of which depend on the cue. If the vortex indeed reflects the strength of low-level sensory representations that are driven by the cue's salience, then, consistent with previous demonstrations of attentional/oculomotor capture (*Theeuwes, 1991*; *Theeuwes, 1992*; *Theeuwes, 1994*), it should become weaker when the salience of the cue is reduced. To investigate this, our participants performed the compelled antisaccade task with cues of three luminance levels, high (data shown in *Figure 2*), medium, and low (Materials and methods). The three cues were the same for all participants and were randomly interleaved during the experiment. Because the faintest cue was chosen to be slightly above the detection threshold, we expected it to yield a much shallower vortex.

The expectation for the later rise in the tachometric curve was less clear. If the rise is a direct reflexion of the cognitive process that remaps the spatial location of the cue and programs an antisaccade, then its steepness must depend, at least in part, on the speed and the variability of this process. So, if weaker sensory signals are generally processed more slowly or with higher variance, then the tachometric curve should rise more gradually as luminance decreases.

The experimental results showed that both the timing and depth of the vortex depend strongly on cue luminance. For the data pooled from all participants (*Figure 3a*), as luminance decreases from high (bright green points) to low (dark green points), the vortex shifts to the right by about 50 ms (the minimum point shifts from rPT = 111 ± 1.3 ms [SE from bootstrap] to rPT = 162 ± 6.0 ms), suggesting that the time needed to detect the cue increases accordingly. The vortex also becomes much less deep (the minimum fraction correct goes from 0.03 ± 0.006 to 0.32 ± 0.026). These findings are consistent with the expected weakening of (involuntary) attentional capture.

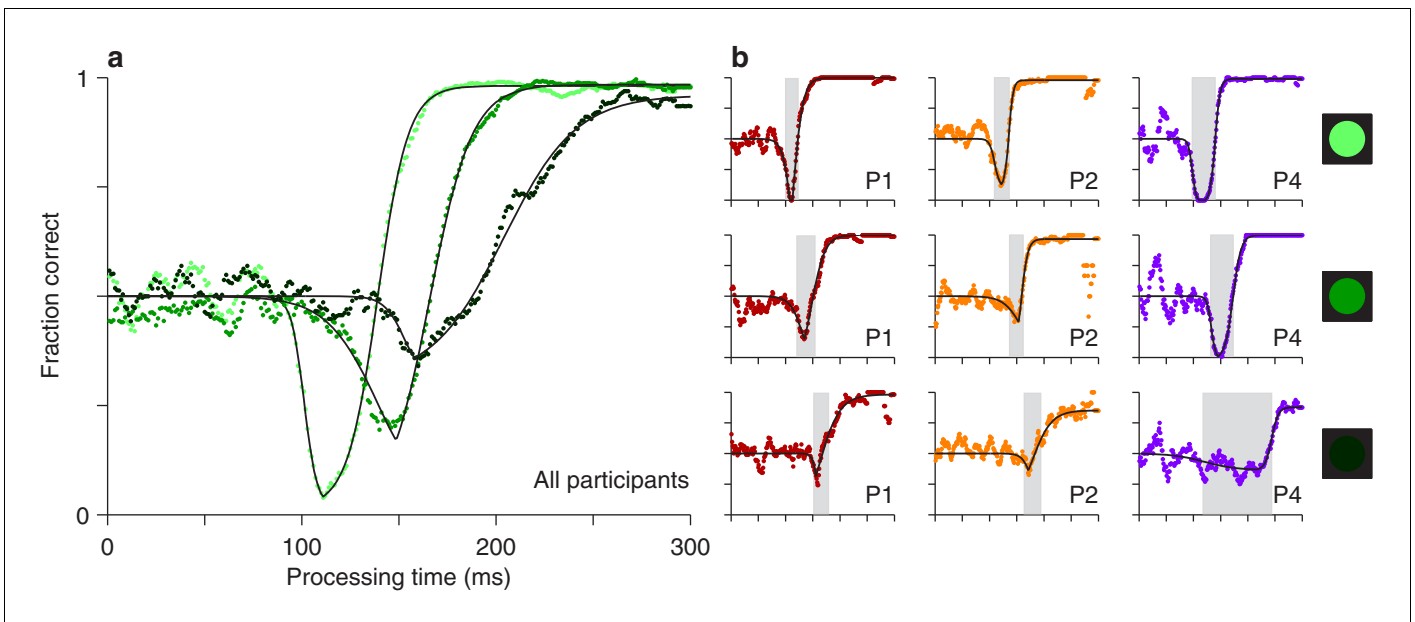

**Figure 3.** Perceptual performance varies as a function of cue luminance. (**a**) Tachometric curves for trials in which the cue had high, medium, and low luminance (indicated by bright, grayish, and dark green points, respectively). Results are for the pooled data from all participants. The vortex shifts to the right and becomes less deep as luminance decreases. (**b**) Tachometric curves from three individual participants at each cue luminance level, high, medium and low, as indicated by the icons. Gray shades demarcate the vortex of each curve. In each panel, colored points are experimental results and black lines are continuous fits to the data.

DOI: https://doi.org/10.7554/eLife.46359.006

We also found that, as luminance decreases, the rise of the tachometric curve becomes significantly less steep ($p < 0.0001$ for all differences in maximum slope between luminance conditions, from bootstrap; see *Figure 4—figure supplement 1*), consistent with the notion that the voluntary remapping of the cue location proceeds more slowly or less reliably as the cue becomes more dim. Thus, in addition to strongly determining the initial, bottom-up response to the cue, luminance probably also impacts the top-down process at work in the task.

Qualitatively similar dependencies on cue luminance were observed in each participant's data set (*Figure 3b*), but reliable differences across individuals became evident when the effects were evaluated quantitatively. For any given tachometric curve, quantification was achieved by fitting the empirical data (*Figure 3*, colored data points) with a continuous analytical function (*Figure 3*, black traces; *Equation 2*) and measuring several features from the fitted curve (Materials and methods). We present results for three such features that were particularly reliable given the size of our samples (for additional features, see *Figure 4—figure supplement 1*). The first one is the average value of the tachometric curve for rPTs between 0 and 250 ms, which we refer to as the mean perceptual accuracy (*Figure 4a*). The second feature is the rPT at which the tachometric curve reaches its minimum, which we designate as the vortex time (*Figure 4b*). And the third feature is the rPT at which the rising part of the tachometric curve is halfway between its minimum and maximum values, which we call the endogenous response centerpoint, or just the centerpoint of the curve, for brevity (*Figures 2* and *3b*, right border of gray shades; *Figure 4c*). These quantities are partially related; the centerpoint, which measures how soon the participant can escape the vortex, is independent of the vortex time (partial Spearman correlation $\rho = 0.34$, $p = 0.2$; Materials and methods), but is strongly anti-correlated with perceptual accuracy ($\rho = -0.85$, $p = 10^{-5}$). Notably, the separation between the 'best' and the 'worst' participant within a given luminance condition is statistically large, particularly for the mean perceptual accuracy and the centerpoint of the curve (*Figure 4a,c*; note little overlap between 95% confidence intervals for bars of same color). The observed effects of cue luminance are highly consistent across participants (*Figure 3b*), but the quantitative details reveal idiosyncratic variations that distinguish one individual from another (*Figure 4a,c*; see below). This is significant because cognitive tasks generally produce robust differences either between experimental conditions/treatments or between individuals, but not both (*Borsboom et al., 2009*; *Hedge et al., 2018*).

## Individual differences in perceptual and overall performance

Antisaccade performance is often quantified using the mean accuracy. This assumes that the overall success rate in the task directly reflects the degree to which voluntary control can override the involuntary urge to look at a salient stimulus. But is this assumption correct? Answering this critical question is generally difficult because doing so requires access to an independent assessment of perceptual performance — but that is precisely what the tachometric curve affords. To investigate the relationship between traditional antisaccade performance measures and perceptual capacity, we examined their natural variations accross individual participants.

First, we computed the correlation between two variables (Materials and methods), the average perceptual accuracy (mean value of the tachometric curve), and the average observed accuracy (mean fraction of correct choices). We found that, even though both quantites tend to increase with higher luminance, suggesting a positive correlation, they are, in fact, uncorrelated (*Figure 5a*). The rank of a given participant based on one measure is not predictive of his or her rank based on the other.

This may seem surprising. Logic dictates that better perception should translate into better performance — but critically, this is contingent on everything else being equal. The paradox arises because the mean RT also varies across participants, and the two accuracy measures relate to it in opposite ways. The average observed accuracy demonstrates a strong speed-accuracy tradeoff, that is, slower participants are correct more often (*Figure 5b*). In contrast, the mean perceptual accuracy demonstrates a weaker opposite trend, that is, those participants that exhibit high perceptual ability also tend to respond more quickly (*Figure 5c*). The results are nearly identical when perceptual performance is quantified with the endogenous response centerpoint (*Figure 5—figure supplement 1*).

This stark divergence did not arise because our urgent task produced more or different errors, but rather because, unlike the mean accuracy, the tachometric curve is a true metric of perceptual processing. This curve is highly sensitive to the properties of the visual stimuli that must be judged

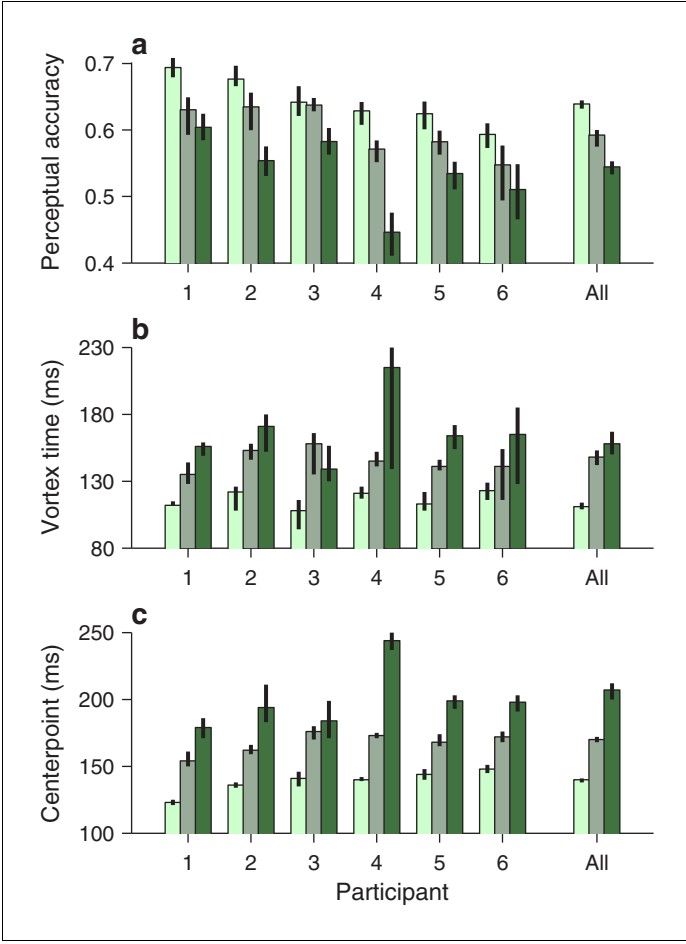

**Figure 4.** Perceptual performance quantified across participants and luminance conditions. Each panel shows one particular quantity derived from the fitted tachometric curves, with results sorted by participant (x axes) and luminance level, high (bright green), medium (grayish green), and low (dark green). Error bars indicate 95% confidence intervals obtained by bootstrapping. (**a**) Mean perceptual accuracy, calculated as the average value of the fitted tachometric curve for rPTs between 0 and 250 ms. (**b**) Vortex time, calculated as the rPT at which the minimum of the fitted tachometric curve is found. (**c**) Endogenous response centerpoint, equal to the rPT at which the rise of the fitted tachometric curve is halfway between its minimum and maximum values.

DOI: https://doi.org/10.7554/eLife.46359.007

The following figure supplements are available for figure 4:

**Figure supplement 1.** Additional quantities that characterize perceptual performance across participants and cue conditions.
DOI: https://doi.org/10.7554/eLife.46359.008

**Figure supplement 2.** Simulated perceptual performance quantified in the same way as the experimental data.
DOI: https://doi.org/10.7554/eLife.46359.009

(such as luminance, in this case), and at the same time, when such properties are fixed, it is largely impervious to manipulations that substantially alter the RT (*Stanford et al., 2010*; *Shankar et al., 2011*; *Salinas et al., 2014*; *Scerra et al., 2019*; for evidence that is specific to the compelled anti-saccade task, see *Figure 5—figure supplement 2*). In contrast, the mean observed accuracy depends not only on the shape of the tachometric curve, but also on two factors, unrelated to perception, that determine which parts of the curve are sampled during an experiment, the gap values used and the subject's urgency. For instance, when only a zero gap is used, most rPTs are beyond the vortex range (*Figure 2—figure supplement 1g,h*), where differences between participants reflect mainly their asymptotic performance levels, that is, their lapse rates. But even when a wide range of gaps is used (as in *Figure 5*), participants that tend to respond quickly (short RTs) will

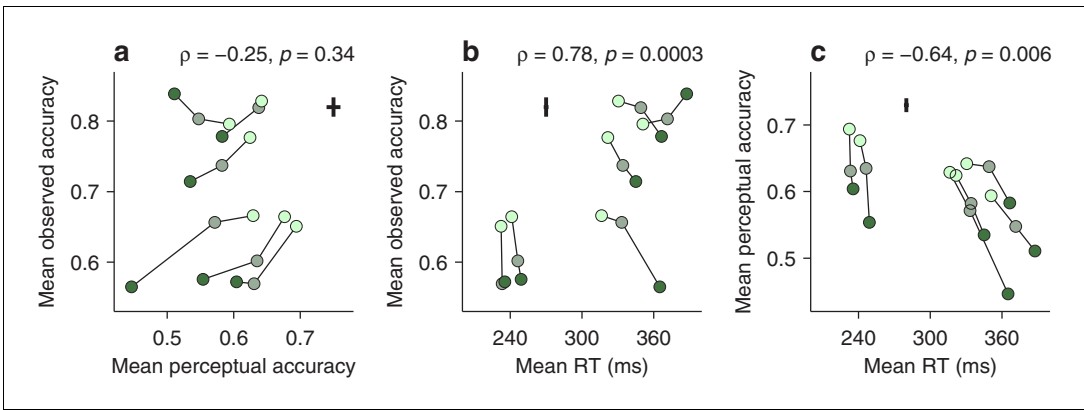

**Figure 5.** Dissociation between perceptual capacity and overall task performance. In each panel, the data from each participant (joined by lines) are shown for trials of high, medium, and low luminance cues (bright, grayish, and dark green points, respectively). Crosses indicate the typical (median) uncertainty (2 SEs) associated with the measurement in each direction. Partial Spearman correlations between values on the x and y axes are indicated, along with significance (Materials and methods). The partial correlation eliminates the association due exclusively to luminance. (a) Mean observed accuracy versus mean perceptual accuracy. (b) Mean observed accuracy versus mean RT. Average RT data include both correct and incorrect trials. (c) Mean perceptual accuracy versus mean RT.
DOI: https://doi.org/10.7554/eLife.46359.010

The following figure supplements are available for figure 5:

**Figure supplement 1.** Dissociation between perceptual capacity and overall task performance based on the curve centerpoint.
DOI: https://doi.org/10.7554/eLife.46359.011

**Figure supplement 2.** Decoupling perceptual and motor performance.
DOI: https://doi.org/10.7554/eLife.46359.012

**Figure supplement 3.** Dissociation between perceptual capacity and overall task performance as seen with the accelerated race-to-threshold model.
DOI: https://doi.org/10.7554/eLife.46359.013

**Figure supplement 4.** Model parameters that characterize individual perceptual performance.
DOI: https://doi.org/10.7554/eLife.46359.014

generally produce short rPTs and sample more densely the left side of their curves, whereas participants that tend to respond slowly (long RTs) will generally produce long rPTs and sample more densely the right side of their curves. This is the source of the speed-accuracy tradeoff found here (*Figure 5b*). As a result, the mean accuracy provides scarcely any information about the ability of an individual to prevent a captured saccade relative to that of others.

## A comprehensive account of antisaccade behavior based on motor competition

We developed a physiologically feasible model (Materials and methods) to explore two mechanistic hypotheses about the neural origin of the vortex. This model is a variant of one that replicates both behavioral performance and choice-related neuronal activity (in the FEF) in an urgent, two-alternative, color discrimination task (*Stanford et al., 2010*; *Shankar et al., 2011*; *Costello et al., 2013*; *Seideman et al., 2018*). As in that case, the current model considers two variables, $r_L$ and $r_R$, that represent oculomotor responses favoring saccades toward left and right locations (*Figure 6*, black and red traces). These motor plans compete with each other such that the first one to reach a fixed threshold level (*Figure 6*, dashed lines) determines the choice: a left saccade if $r_L$ reaches threshold first, or a right saccade if $r_R$ reaches threshold first. In each trial, after the go signal, $r_L$ and $r_R$ start increasing with randomly drawn build-up rates. The build-up process is likely to end in a random choice (i.e. a guess; *Figure 6c*) when one of the initial rates is high and/or the gap is long, but otherwise, time permitting, the cue signal modifies the ongoing motor plans (*Figure 6a,b*). Specifically, once the target has been identified, the plan toward it (correct) is accelerated and the other one, toward the opposite, incorrect location, is decelerated (*Figure 6a*, note acceleration of black trace

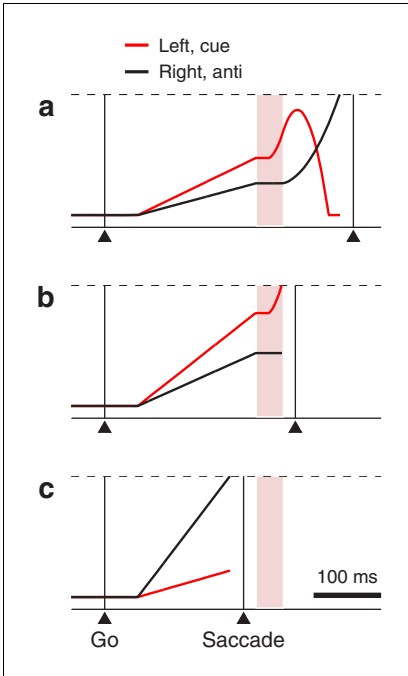

**Figure 6.** Three representative single trials of the race-to-threshold model. Traces show motor plans $r_L$ toward the left (red) and $r_R$ toward the right (black) as functions of time. Because in these examples the cue is assumed to be on the left, these variables also correspond to motor plans toward the cue (red) and the anti location (black), respectively. After the exogenous response interval (ERI, pink shade), the former (incorrect) plan decelerates and the latter (correct) plan accelerates. A saccade is triggered a short efferent delay after activity reaches threshold (dashed lines). (a) A correct, long-rPT trial; that is, an informed choice (RT = 369 ms, rPT = 219 ms). (b) An incorrect trial with rPT within the vortex; that is, a captured saccade (RT = 283 ms, rPT = 133 ms). (c) A correct, short-rPT trial; that is, a correct guess (RT = 206 ms, rPT = 56 ms). In all examples, the gap is 150 ms. The influence of the cue depends on its timing relative to the ongoing motor activity.
DOI: https://doi.org/10.7554/eLife.46359.015

and deceleration of red trace after shaded interval). This corresponds to the cue content, interpreted according to task rules, informing the correct choice.

To adapt this 'accelerated race-to-threshold' model to the compelled antisaccade task, we introduced one crucial, task-specific assumption: that the competition is biased in favor of the cue location during a period of time that we refer to as the exogenous response interval, or ERI (*Figure 6*, pink shades). During the ERI, the cue has already been detected by the circuit but not yet interpreted as 'opposite to the target' (so it cannot yet drive the endogenous acceleration and deceleration described above). We consider two possible mechanisms by which, during the ERI, the detection of the cue may lead to exogenous attentional/oculomotor capture: (1) it could halt or suppress the ongoing plan toward the anti location (*Figure 6a,b*, black traces during pink interval) or (2) it could transiently accelerate the ongoing plan toward the cue location (*Figure 6a,b*, red traces during pink interval). These alternatives are not mutually exclusive. The former is consistent with evidence that salient, abrupt-onset stimuli reflexively interrupt ongoing saccade plans (*Dorris et al., 2007*; *Bompas and Sumner, 2011*; *Hafed and Ignashchenkova, 2013*; *Buonocore et al., 2017*; *Salinas and Stanford, 2018*), whereas the latter is consistent with the short-latency, stimulus-driven activation of visually responsive neurons in oculomotor areas (*Gottlieb and Goldberg, 1999*; *Bisley et al., 2004*; *Thompson et al., 2005*; *Ipata et al., 2006*; *Marino et al., 2015*; *Joiner et al., 2017*; *White et al., 2017*; *Chen et al., 2018*).

We found that, to reproduce the psychophysical data accurately, both mechanisms were necessary. To see why, first note that the tachometric curve, which refers to the proportion of correct choices in each rPT bin, can be expressed as a ratio,

$$C(\mathrm{rPT}) = \frac{f_C(\mathrm{rPT})}{f_C(\mathrm{rPT}) + f_I(\mathrm{rPT})} \quad (1)$$

where $f_C(\mathrm{rPT})$ and $f_I(\mathrm{rPT})$ describe the frequencies of correct and incorrect choices at each rPT, that is, they are the rPT distributions for correct and incorrect trials (normalized by the same factor; Materials and methods). Each of these distributions demonstrates a distinct feature: $f_C$ has a dip (green shades in *Figure 7*, third row), whereas $f_I$ has a peak (red shades in *Figure 7*, bottom row). Both features contribute to the vortex, as dictated by the above expression. The critical mechanistic observation is that acceleration of the motor plan toward the cue during the ERI accounts for the peak in $f_I$ (*Figure 7a*), whereas interruption of the competing motor plan away from the cue produces the dip in $f_C$ (*Figure 7b*). Thus, when the model was implemented with either one of the mechanisms alone, it failed to replicate the experimental feature associated with the other (*Figure 7*, bottom two rows, compare black traces in a vs. b). However, with the two mechanisms acting

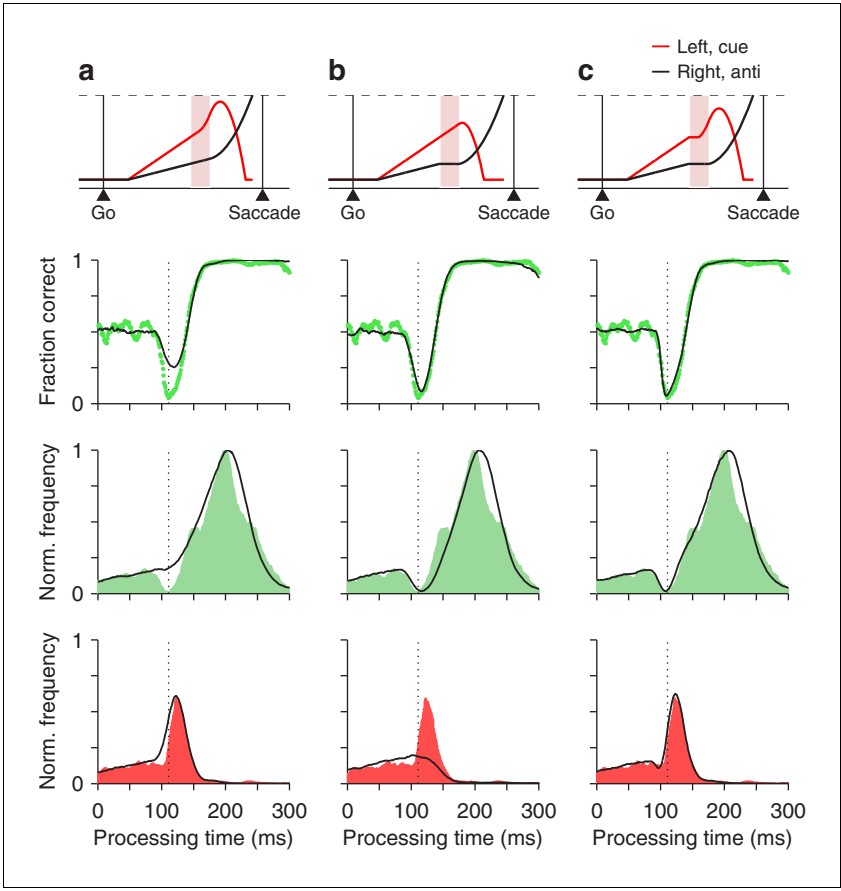

**Figure 7.** Contributions of two distinct neural mechanisms to attentional/oculomotor capture. *Top row*: representative single, long-rPT trials from the model. *Second row*: tachometric curves, simulated (black traces) and experimental (green dots). *Third row*: rPT distributions for correct trials ($f_C$), simulated (black traces) and experimental (green shades). *Bottom row*: rPT distributions for incorrect trials ($f_I$), simulated (black traces) and experimental (red shades). (**a**) Results from a restricted version of the model in which, during the ERI, the motor plan toward the cue accelerates but the plan toward the anti location keeps advancing, unperturbed. (**b**) Results from another restricted version of the model in which, during the ERI, the motor plan toward the anti location halts but that toward the cue keeps advancing, unperturbed. (**c**) Results from the full-blown model, in which, during the ERI, the cue plan accelerates and the anti plan halts. For each model variant, results were obtained with the parameter values that minimized the error between the model and the pooled experimental data in the high luminance condition (Materials and methods).

DOI: https://doi.org/10.7554/eLife.46359.016

simultaneously, in coordination, the model reproduced the full data set in quantitative detail (*Figures 7c* and *8*).

First, for the data pooled across participants, the model fitted the tachometric curve (*Figure 8a*) and the rPT distributions for correct and incorrect responses (*Figure 8b*). Second, for individual gap conditions, the model matched the variations in mean success rate and mean RT (*Figure 8—figure supplement 1*), but more importantly, it reproduced the shapes of the RT distributions for correct and incorrect choices, which were typically bimodal (*Figure 8c*). Third, the model accurately captured all the dependencies on luminance (*Figure 8*, compare results across columns). Importantly, in doing so, the values of the parameters that correspond to pure motor performance (the distribution of initial build-up rates for $r_L$ and $r_R$, and the distribution of afferent delays associated with the go signal) were the same across luminance conditions (*Table 1*), in correspondence with the fact that all trials proceeded identically up to cue presentation, and that trials with different gap and cue luminance were interleaved during the experiment. And fourth, the model also fitted the (noisier) data from individual participants, even though they showed large, idiosyncratic variations in motor

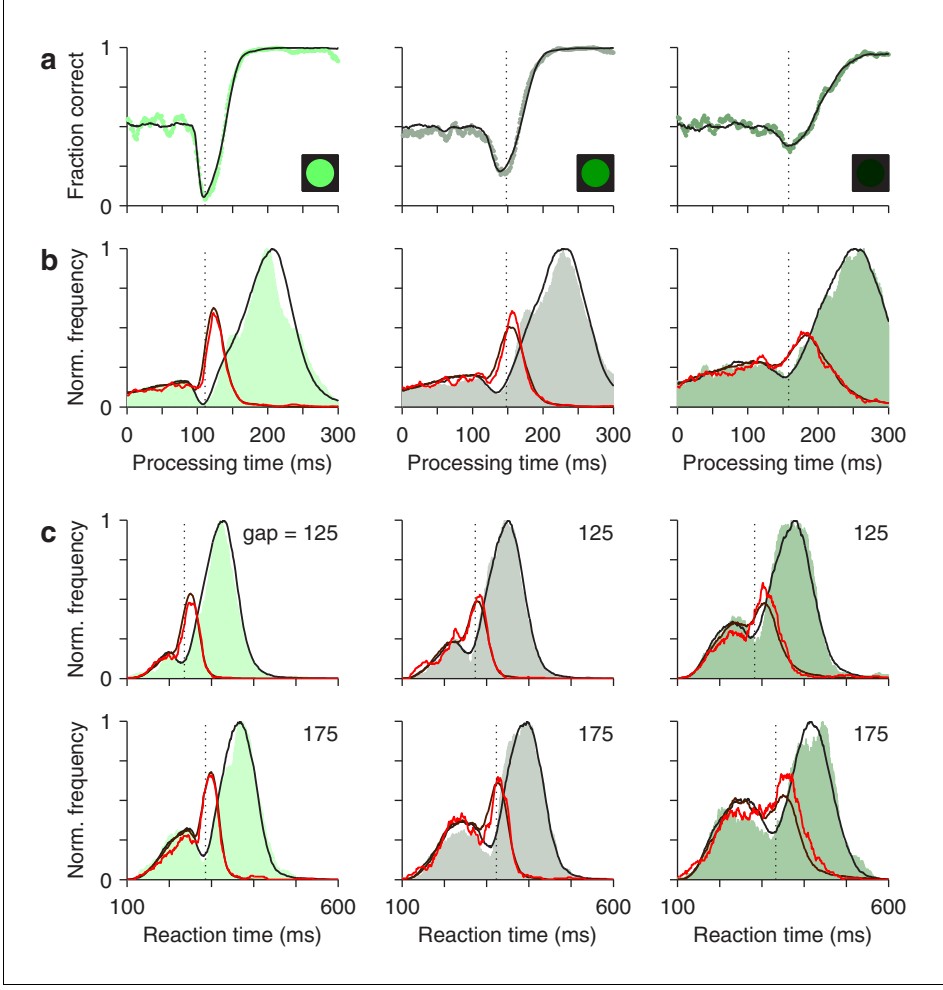

**Figure 8.** The race-to-threshold model accounts for antisaccade performance. (**a**) Tachometric curves for high (left), medium (middle), and low (right) luminance cues. Continuous lines are model results. (**b**) Processing time distributions for correct (shades) and incorrect trials (red traces) at each luminance level. Overlaid traces (black and dark red) are corresponding model results. (**c**) RT distributions for correct (shades) and incorrect trials (red traces) at individual gap values (125 and 175 ms) for each luminance level. Overlaid traces (black and dark red) are model results. Dotted vertical lines mark the rPT at which the vortex reaches its minimum point (vortex time). All empirical data are pooled across participants.

DOI: https://doi.org/10.7554/eLife.46359.017

The following figure supplements are available for figure 8:

**Figure supplement 1.** The race-to-threshold model reproduces average performance in individual gap conditions.
DOI: https://doi.org/10.7554/eLife.46359.018

**Figure supplement 2.** The accelerated race-to-threshold model individually fitted to the data of participants 1 (top two rows), 2 (middle rows), and 3 (bottom two rows), as indicated on the right.
DOI: https://doi.org/10.7554/eLife.46359.019

**Figure supplement 3.** The accelerated race-to-threshold model individually fitted to the data of participants 4 (top two rows), 5 (middle rows), and 6 (bottom two rows).
DOI: https://doi.org/10.7554/eLife.46359.020

performance, as well as in the dips and peaks of their rPT distributions (*Figure 8—figure supplements 2* and *3*). For all comparisons across participants, the empirical (*Figures 4* and *5*) and simulated results (*Figure 4—figure supplement 2*; *Figure 5—figure supplement 3*) were nearly indistinguishable.

**Table 1.** Parameters of the race-to-threshold model for the pooled data.
Build-up rates are in AU ms$^{-1}$, times are in ms, and acceleration and deceleration are in AU ms$^{-2}$.

| Lum | $\mu_b$ | $\sigma_b$ | $\rho_b$ | $\mu^{\mathrm{aff}}_{\mathrm{GO}}$ | $\sigma^{\mathrm{aff}}_{\mathrm{GO}}$ | $\mu^{\mathrm{aff}}_{\mathrm{CUE}}$ | $\sigma^{\mathrm{aff}}_{\mathrm{CUE}}$ | $\mu_{\mathrm{ERI}}$ | $\sigma_{\mathrm{ERI}}$ | $g_{\mathrm{ERI}}$ | $\Delta_{\mathrm{ERI}}$ | $a_{\mathrm{EX}}$ | $d_{\mathrm{END}}$ | $a_{\mathrm{END}}$ | $\lambda$ |
|---|---|---|---|---|---|---|---|---|---|---|---|---|---|---|---|
| High | 1.4 | 3.74 | −0.95 | 51 | 36 | 76 | 5 | 24 | 4 | 0 | 10 | 0.96 | −0.7 | 0.17 | 0.02 |
| Medium | 1.4 | 3.74 | −0.95 | 51 | 36 | 104 | 13 | 24 | 3 | 0 | 14 | 1.15 | −0.54 | 0.17 | 0.02 |
| Low | 1.4 | 3.74 | −0.95 | 51 | 36 | 126 | 19 | 24 | 10 | 0 | 14 | 0.58 | −0.29 | 0.14 | 0.1 |

DOI: https://doi.org/10.7554/eLife.46359.021

Mechanistically, the best-fitting parameter values of the model (*Table 1*) provide further insight about the crucial element that gives rise to the vortex — the exogenous bias during the ERI (*Figure 6*, pink shades). Consider the following values based on the pooled data. According to the model, the onset of the ERI, which corresponds to the time at which the cue is detected, is highly sensitive to luminance. For the high, medium, and low conditions, the oculomotor circuitry detects the cue 76 ± 5 ms (mean ± SD for simulated trials), 104 ± 13 ms, and 126 ± 19 ms after its presentation. This variation from high to low luminance (50 ms) corresponds closely with the rightward shift of the vortex observed experimentally (51 ms; *Figure 3a*) and is consistent with the ubiquitous dependence of visual response latency on luminance and contrast (*Purushothaman et al., 1998*; *Bisley et al., 2004*; *van Rossum et al., 2008*; *White et al., 2008*; *Oram, 2010*; *Marino et al., 2015*). Remarkably, the ERI lasts only 24 ms (on average) in all three conditions, and the exogenous acceleration of the plan toward the cue occurs only during the last 14 ms (high luminance), or only during the last 10 ms (medium and low luminance); before that, the plan toward the cue halts just like its counterpart toward the anti location (*Figure 6a,b*, note that red trace is initially flat during shaded interval). The model suggests that the exogenous acceleration favoring the cue location is very brief but very powerful, which explains why the left edge of the vortex can be so steep.

Finally, the parameter values (*Supplementary file 1*) also point to specific neural mechanisms that likely underlie the individual differences in perceptual capacity. In general, identifying those mechanisms is complicated because their variations across (random) participants and across (controllable) experimental conditions are not necessarily correlated (*Borsboom et al., 2009*; *Hedge et al., 2018*). The cue latency discussed above is a perfect example: it demonstrates (via parameter $\mu^{\mathrm{aff}}_{\mathrm{CUE}}$) a strong, consistent dependence on luminance in each participant's data set, and yet, for a given luminance level, it is not predictive of individual perceptual accuracy (*Figure 5—figure supplement 4a*). By contrast, we hypothesize that the magnitudes of the exogenous and endogenous acceleration (via parameters $a_{\mathrm{EX}}$ and $a_{\mathrm{END}}$) are major sources of individual variation, because although they have weaker dependencies on luminance, they are reliable predictors of perceptual accuracy (*Figure 5—figure supplement 4b–d*).

## Discussion

By making the antisaccade task urgent, focusing on processing time (instead of RT), and developing a mechanistic model that is firmly grounded on the neurophysiology of saccadic choices, we were able to resolve the opposing influences of endogenous and exogenous mechanisms on the oculomotor response with unprecedented sharpness. Our findings suggest that, whether overt or covert, the capture of attention by a salient stimulus corresponds to specific changes in the firing of saccade-related neurons: activity that is spatially congruent with the stimulus is accelerated, whereas activity that is spatially incongruent is halted or suppressed. This fast, reflexive bias *is* the capture.

The results make four contributions. First, they identify concrete ways in which endogenous and exogenous mechanisms act on the oculomotor circuitry — namely, via acceleration, deceleration, and halting of rising firing rates — together with their unique behavioral signatures (dips and peaks in rPT and RT distributions). Second, they characterize the time scales of those mechanisms (a few tens of ms) as well as their dependencies on luminance. Third, they clearly parse the motor (RT) and perceptual (tachometric curve) contributions to antisaccade performance, thus removing the pervasive confounds caused by the speed-accuracy tradeoff. And fourth, they relate variations in motor, perceptual, and cognitive mechanisms to individual differences in performance. In particular, all

participants exhibited a vortex, but different vortices resulted from different combinations of exogenous and endogenous mechanisms (compare rPT distributions for P1 vs. P4 in *Figure 8—figure supplements 2* and *3*) — a degeneracy in neural function that is to be expected based on that found in more reduced circuits (*Marder et al., 2015*).

## Antisaccade performance in relation to cognitive conflict

By design, the antisaccade task creates a conflict between exogenous and endogenous mechanisms, the former driven by the saliency of the cue and the latter by task instructions followed willfully. Other tasks (e.g. *Kim and Cave, 1999*), most notably the singleton-distracter task employed by Theeuwes and colleagues (*Theeuwes, 1991*; *Theeuwes, 1992*; *Theeuwes, 1994*; *Theeuwes et al., 1998*; *Theeuwes et al., 1999*; *Nissens et al., 2017*), have also revealed such conflict in the form of attentional or oculomotor capture, but its manifestation in those cases consists primarily of small variations (~ tens of ms) in RT around mean values that are much longer (>> 250 ms) than typical inter-saccadic intervals, and the results can only provide a crude estimate of the underlying temporal dynamics (*Mulckhuyse et al., 2008*; see also *Markowitz et al., 2011*). Those tasks also require more complex visual displays with multiple items, and a secondary discrimination to serve as a probe of the effect. In principle, a minimalistic task typifying such an essential phenomenon would be extremely useful; it could serve to determine the neural correlates of volitional versus reflexive action, or pinpoint the consequences of disease on specific cognitive abilities, for example.

Numerous studies based on the traditional antisaccade task have, in fact, reported large differences in overall performance between distinct populations of participants (*Guitton et al., 1985*; *Klein and Foerster, 2001*; *Munoz et al., 2003*; *Condy et al., 2007*; *Hakvoort Schwerdtfeger et al., 2012*; *Antoniades et al., 2015*) — but do those differences relate to the conflict at the heart of the task? This is unclear. While a distinction between fast and slow errors has been drawn based on theoretical considerations (*Lo and Wang, 2016*), the tachometric curve reveals three types of error: fast, incorrect guesses (rPT $\leq$ 100 ms), saccades captured by the cue (vortex), and lapses (rPT $\geq$ 200 ms), which probably depend on distinct cognitive processes or states that vary over long time scales (*Harris and Thiele, 2011*; *Lo and Wang, 2016*; *Nir et al., 2017*). When antisaccade performance is evaluated via the mean accuracy, the most common metric, the three error types are combined in proportions that are unpredictable, because they depend on each participant's urgency (how quickly they tend to respond; *Figure 5b*) and on the gap values used in the experiment. In particular, when the cue is presented before or simultaneously with the go signal, most rPTs are sampled in the asymptotic performance range, where most errors are lapses (*Figure 2—figure supplement 1*). The captured saccades are the essential manifestation of the conflict, but they cannot be reliably quantified unless performance is tracked with high temporal resolution.

## Oculomotor capture as a perceptuo-motor phenomenon

Insight about the visuo-motor interactions that determine the shape of the tachometric curve can be gleaned by realizing that, for each trial, the rPT conveys information not only about how much time was available for perceptual deliberation, but also about the state of the motor build-up at the time when the cue was detected by the oculomotor circuitry. Consider what must happen for a saccade to be triggered at rPT = 111 ms, when the capture is nearly certain: at the moment that the cue is detected (left edge of pink interval in *Figure 6b*), the motor activity toward the cue location must be just below threshold, so that the exogenous drive can reliably propel it past threshold. This can happen in many ways, such as when the motor plans build-up slowly and the gap is long, or when the build-up is fast and the gap is short — but whatever the build-up history, an rPT of 111 ms corresponds to the requisite level of subthreshold activity. The same is true for other parts of the tachometric curve. Short rPTs (guesses) correspond to insufficient perceptual deliberation and to activity that exceeded threshold before the cue was detected (*Figure 6c*), whereas long rPTs (informed choices) correspond to successful deliberation guiding activity that was still far from threshold at the time of cue detection (*Figure 6a*). Captured saccades are reliably found within a narrow range of rPTs because their expression requires certain combinations of sensory (cue exposure) and motor (degree of build-up) conditions to be met.

This explanation simply recounts what the model does, so it is worth discussing how our model is different from previous ones, and why we think it is largely credible. Previous models of antisaccade

performance (e.g. *Wiecki and Frank, 2013*; *Lo and Wang, 2016*; *Aponte et al., 2017*) applied to non-urgent conditions, so they provide limited insight about the shape of the tachometric curve. Furthermore, such models were concerned with the neural basis of inhibitory control more generally, so they involve, explicitly or implicitly, multiple brain areas; for example, one for producing motor responses and another for inhibiting the reflexive movement toward the cue. In contrast, our race-to-threshold model is agnostic as to where the relevant perceptual or control signals come from, or how they are computed; it simply deals with their dynamical impact (i.e. acceleration, deceleration, halting) on the developing motor activity that must ultimately communicate the urgent choice. As such, the fast variations of the model firing rates are meant to be directly comparable to those of saccade-related oculomotor neurons (in FEF, and perhaps SC).

Indeed, previous single-neuron studies in monkeys support key elements of our modeling framework. First, the initial level of motor activity contributes as proposed: during urgent saccadic choices, the build-up of activity in FEF starts after the go signal, regardless of when the cue information arrives (*Stanford et al., 2010*; *Costello et al., 2013*), and during antisaccade performance, the ongoing activity in FEF and SC is higher before erroneous saccades toward the cue than before correct antisaccades (*Everling et al., 1998*; *Everling and Munoz, 2000*). Second, in an urgent color discrimination task, perceptual information does produce acceleration and deceleration of motor activity in an rPT-dependent way (*Stanford et al., 2010*; *Costello et al., 2013*). Third, the timing of the vortex and its dependence on luminance parallel those of visual bursts in the oculomotor system (*Gottlieb and Goldberg, 1999*; *Bisley et al., 2004*; *Thompson et al., 2005*; *Ipata et al., 2006*; *Joiner et al., 2017*; *White et al., 2017*; *Chen et al., 2018*). In particular, the captured saccades in our experiment resemble so-called 'express saccades' in many ways: both result in movements triggered after 100 ms or less of stimulus viewing time; both are more likely with higher luminance; and both are facilitated by the early removal of the fixation requirement, such that a visually evoked response is superimposed on advancing motor activity (*Paré and Munoz, 1996*; *Dorris et al., 1997*; *Marino et al., 2015*). And fourth, the sudden presentation of a salient distracter stimulus has a robust impact on a developing saccade plan, with the effect depending strongly on their spatial congruence: when the saccade target is diametrically opposite to the stimulus, there is ample evidence (reviewed by *Salinas and Stanford, 2018*) indicating that the developing plan is transiently halted or suppressed, whereas when the saccade target is near the abrupt-onset stimulus, the developing plan is boosted (*Dorris et al., 2007*; *Edelman and Xu, 2009*; *White et al., 2013*; *Marino et al., 2015*). These observations are consistent with the exogenous and endogenous mechanisms implemented by the model.

## Coupling between spatial attention and saccade planning

Our results are pertinent to a mechanistic question that is central to the 'premotor theory' of attention: to what degree is the neural substrate of the deployment of spatial attention the same as that of saccade planning? There is strong evidence that the rise in oculomotor activity associated with planning a saccade inevitably implies that attentional resources are at least partially allocated to the intended saccade endpoint (*Kowler et al., 1995*; *Deubel and Schneider, 1996*; *Moore and Fallah, 2001*; *Godijn and Theeuwes, 2003*; *Moore and Armstrong, 2003*; *Cavanaugh and Wurtz, 2004*; *Steinmetz and Moore, 2014*; *Klapetek et al., 2016*). The converse relationship — that is, whether the covert deployment of spatial attention must be accompanied by saccade planning — has been more contentious (*Juan et al., 2004*; *Thompson et al., 2005*). It appears, however, that the hypothesized motor plan associated with attentional allocation is just very difficult to observe when fixation must be actively maintained (*Belopolsky and Theeuwes, 2012*). During fixation, such a plan may manifest only as a subtle increase in baseline activity, rather than via the more typical steady rise in firing rate (*Hauser et al., 2018*), but it can be uncovered through experimental manipulations (*Theeuwes et al., 1998*; *Theeuwes et al., 1999*; *Katnani and Gandhi, 2013*; *Nissens et al., 2017*), and is evident in microsaccades (*Chen et al., 2015*; *Lowet et al., 2018*).

Our results are consistent with the idea that attentional and oculomotor capture are different behavioral manifestations of the same underlying neuronal dynamics (in, say, the FEF or SC). When a salient cue is detected, a bias favoring a motor plan toward its location is always generated (with the bias consisting of acceleration of the plan toward the cue and halting of any plans away from it). However, the impact of the exogenous biasing signal depends on the current state of the oculomotor circuitry. When the motor activity congruent with the cue location is far from threshold and is

competing with other developing motor plans, the bias corresponds to the covert (and transient) deployment of attention to the cue. In contrast, when that activity has already developed to a substantial degree, the bias can quickly propel it past threshold. Then, the exogenous attraction of the cue becomes observable as an overt, captured saccade.

The detailed mechanistic framework presented here should provide ample opportunity to test these ideas in future experiments.

## Materials and methods

### Subjects and setup

Experimental subjects were six healthy human volunteers, two male and four female, ages 21–30. They were recruited from the Wake Forest School of Medicine and Wake Forest University communities. All had normal or corrected-to-normal vision. All participants provided informed written consent before the experiment. All experimental procedures were conducted with the approval of the Institutional Review Board (IRB) of Wake Forest School of Medicine.

The experiments took place in a semi-dark room. The participants sat on an adjustable chair, with their chin and forehead supported, facing a VIEWPixx LED monitor (VPixx Technologies Inc, Saint Bruno, Quebec, Canada; 1920 × 1200 screen resolution, 120 Hz refresh rate, 12 bit color) at a distance of 57 cm. Viewing was binocular. Eye position was recorded using an EyeLink 1000 infrared camera and tracking system (SR Research, Ottawa, Canada) with a sampling rate of 1000 Hz. Stimulus presentation and data collection were controlled using the system's integrated software package (Experiment Builder).

### Behavioral tasks

The sequence of events in the antisaccade task is described in *Figure 1*. The inter-trial interval was 1 s. The gap values used were −200, −100, 0, 75, 100, 125, 150, 175, 200, 250, and 350 ms, where negative numbers correspond to delays in the easy antisaccade task (*Figure 1b*). Thus, compelled and easy, non-urgent trials were interleaved. In each trial, the gap value, cue location (−10° or 10°), and luminance level (see below) were randomly sampled. Auditory feedback was provided at the end of each trial: a beep to indicate that the saccadic response was made within the allowed RT window (450 ms), or no sound if the limit was exceeded. This was independent of the choice. Feedback about the choice itself was unnecessary, as participants easily understood the rules of the task. The task was run in blocks of 150 trials. After 50–150 trials of practice, each participant completed 30 blocks over six experimental sessions (days). Within each session, 2–3 min of rest were allowed between blocks.

The cue was a green circle (0.5° diameter) appearing on a black background. Each participant performed the task with cues of three luminance levels, high (17.6 cd m$^{-2}$), medium (0.35 cd m$^{-2}$), and low (0.22 cd m$^{-2}$). Luminance was measured with a spectrophotometer (i1 Pro 2 from X-Rite, Inc, Grand Rapids, MI). The cues were generated in Adobe Illustrator using the 8-bit RGB vectors [15 168 40], [3 28 7], and [1 12 3]. The lowest luminance was chosen to be close to the detection threshold based on detection curves generated previously for two participants.

### Data analysis

All data analyses were carried out using customized scripts written in Matlab (The MathWorks, Natick, MA). Except where explicitly noted, results are based on the analysis of urgent trials (gap $\geq$ 0) only; that is, easy trials (delay trials with gap < 0) were excluded.

In each trial, the rPT was computed by subtracting the gap value from the RT value recorded in that trial. We refer to this processing time as 'raw' because it includes any afferent or efferent delays in the circuitry (*Stanford et al., 2010*). To compute the tachometric curve and rPT distributions, trials were grouped into rPT bins of 15 ms, with bins shifting every 1 ms. Normalized rPT distributions, $f_C(\text{rPT})$ and $f_I(\text{rPT})$, were obtained by counting the numbers of correct and incorrect trials, respectively, in each rPT bin, and dividing both functions by the same factor. The tachometric curve, which gives the proportion of correct trials in each bin, was then computed using *Equation 1*. For display purposes, the normalization factor used was the maximum value of $f_C$ or $f_I$, whichever was largest, but the factor has no effect on the tachometric curve.

In order to quantify perceptual performance, each tachometric curve was fitted with a continuous analytical function, $v(x)$, which was defined as

$$v(x) = \max(s_L(x), s_R(x), 0) \tag{2}$$

where the maximum function $\max(a, b, c)$ returns $a$, $b$, or $c$, whichever is largest, and $s_L$ and $s_R$ are two sigmoidal curves. These are given by

$$s_L(x) = B + \frac{A_L - B}{1 + \exp(\frac{x - C_L}{D_L})} \tag{3}$$

$$s_R(x) = B + \frac{A_R - B}{1 + \exp(-\frac{x - C_R}{D_R})} \tag{4}$$

where $s_L$ tracks the left (decreasing) side of the tachometric curve and $s_R$ tracks the right (increasing) side. The asymptotic value on the left side was fixed at $A_L = 0.5$, to enforce the constraint that, for very short processing times, performance must be at chance. For any given empirical tachometric curve, the six remaining parameters defining $v$, the coefficients $B$, $A_R$, $C_L$, $C_R$, $D_L$, and $D_R$, were adjusted to minimize the mean absolute error between the experimental and fitting functions. The minimization was done using the Matlab function fminsearch.

Once the best-fitting $v(\mathrm{rPT})$ function for a given tachometric curve was found, we numerically calculated eight quantities, or features, from it: the asymptotic value (equal to $A_R$), the minimum value (vortex depth), the rPT at which the minimum was found (vortex time), the most negative slope, the most positive slope, the rPT for which $v$ is exactly between 0.5 (chance) and the minimum (left edge of the curve), the rPT for which $v$ is exactly between the minimum and the asymptote (the curve's centerpoint), and the average of the curve for rPTs between 0 and 250 ms (mean perceptual accuracy). In *Figures 2* and *3*, the gray shades demarcating the vortex correspond to the interval between the left edge and the centerpoint of each tachometric curve. Confidence intervals for all of these quantities were obtained by bootstrapping (*Davison and Hinkley, 2006*; *Hesterberg, 2014*); that is, by resampling the data with replacement and recalculating all the quantities many times to generate distributions for them. This was done in five steps: (1) resample the original trials with replacement, keeping the original number of contributing trials, (2) recompute the empirical tachometric curve from the resampled trials, (3) fit the new tachometric curve with a continuous $v$ function, (4) recompute the eight characteristic features from the new $v$ function, and (5) repeat steps 1–4 10,000 times to generate distributions for all the features. Reported confidence intervals correspond to the 2.5 and 97.5 percentiles obtained from the bootstrapped distributions.

To quantify the association between average quantities computed for individual participants, such as the mean RT or mean observed accuracy (*Figure 5*), we considered three data points per participant, one for each luminance condition. The strength of association and its significance were calculated with three methods. First we computed the partial Pearson correlation coefficient, which is the standard linear correlation between two variables but controlling for the effect of a third one, luminance in this case. This was implemented via the Matlab function `partialcorr`. We also computed the partial Spearman correlation coefficient, which involves a similar calculation but based on the ranks of the data points. This was using `partialcorr` too. Finally, using the Matlab function `fitlm`, we fitted the data to a linear regression model that also included luminance as a variable. The three methods typically produced similar results. We report those obtained with the partial Spearman correlation, which is denoted as $\rho$, because they were generally the most conservative.

## The accelerated race-to-threshold model

The model for the compelled antisaccade task is a straightforward extension of one developed previously for a two-alternative, urgent, color discrimination task (*Stanford et al., 2010*; *Shankar et al., 2011*; *Costello et al., 2013*; *Seideman et al., 2018*). In that earlier model, both motor plans halt briefly when the relevant cue is detected (i.e. during the ERI); the dynamics are otherwise identical. As explained in the main text, the idea is that two motor plans (in the FEF), represented by firing rate variables $r_L$ and $r_R$, compete with each other to trigger an eye movement with a saccade vector pointing either to the left or to the right. Because the acceleration, deceleration, and halting of these

plans depends on the cue location, it is useful to relabel the two variables as $r_C$ and $r_A$, where the subscripts now refer to the cue and anti locations (keeping in mind that the $C$ and $A$ labels are randomly assigned to left and right directions in each trial). Note, however, that the following description applies identically if the $A$ and $C$ labels are replaced everywhere by $L$ and $R$, respectively, and we assume that the cue appears on the left side.

Over time, the two motor plans advance toward a fixed threshold (equal to 1000 arbitrary units, or AU). If $r_C$ exceeds threshold first, the saccade is incorrect, toward the cue, whereas if $r_A$ exceeds threshold first, the saccade is correct, away from the cue. The saccade is considered to be triggered a short efferent delay (equal to 20 ms) after threshold crossing. The fixed threshold is a reasonable approximation of the triggering mechanism for saccades (**Hauser et al., 2018**).

The two rate variables evolve as follows

$$\begin{aligned} r_C(t+\Delta t) &= r_C(t) + b_C\,\Delta t \\ r_A(t+\Delta t) &= r_A(t) + b_A\,\Delta t \end{aligned} \tag{5}$$

where $b_C$ and $b_A$ are their respective build-up rates and the time step $\Delta t$ is equal to 1 ms. When the build-up rates are constant, the firing rates $r_C$ and $r_A$ increase linearly over time. Periods during which the activity accelerates or decelerates are those during which the build-up rates themselves change steadily, as described below. Any negative $r_C$ and $r_A$ values are reset to zero. Each simulated trial can be subdivided into three epochs with different model dynamics.

Epoch 1: before the ERI. Each trial starts with the two activity variables, $r_C$ and $r_A$, equal to zero. The go signal occurs at $t = 0$, but the two motor plans start building up later, after an afferent delay. This afferent delay is drawn from a Gaussian distribution with mean $\mu_{\mathrm{GO}}^{\mathrm{aff}}$ and SD $\sigma_{\mathrm{GO}}^{\mathrm{aff}}$, where values below 20 ms are excluded. The initial build-up rates, $b_C^0$ and $b_A^0$, are drawn from a two-dimensional Gaussian distribution with mean $\mu_b$, SD $\sigma_b$, and correlation coefficient $\rho_b$. During this epoch, after the initial afferent delay has elapsed, $r_C$ and $r_A$ evolve according to **Equations 5**, with $b_C = b_C^0$ and $b_A = b_A^0$. If during this period one of the motor plans exceeds the threshold, a saccade is produced and the trial ends. Otherwise, the trial continues.

Epoch 2: during the ERI. The start of the ERI corresponds to the time point at which the cue is detected by the model circuit (we stress that this is a local event, and make no claims about the participant's perceptual experience). Cue detection occurs after an afferent delay relative to the time of cue presentation, which is at $t = $ gap. This delay is drawn from a Gaussian distribution with mean $\mu_{\mathrm{CUE}}^{\mathrm{aff}}$ and SD $\sigma_{\mathrm{CUE}}^{\mathrm{aff}}$, where values below 20 ms are excluded. The duration of the ERI also varies normally across trials. It is drawn from a Gaussian distribution with mean $\mu_{\mathrm{ERI}}$ and SD $\sigma_{\mathrm{ERI}}$, with negative values reset to zero. The two motor plans behave differently during the ERI. For the plan toward the anti location, $r_A$, the build-up rate is $b_A = g_{\mathrm{ERI}}\,b_A^0$, where the constant gain factor $g_{\mathrm{ERI}}$ is either zero (i.e. the plan halts) or negative (i.e. the plan is suppressed). This factor was set to zero for the pooled data, but negative values were allowed when fitting the data from individual participants. Whether zero or negative, the build-up rate of the anti plan is the same throughout the whole ERI. In contrast, for the motor plan toward the cue, $r_C$, the build-up rate is $b_C = g_{\mathrm{ERI}}\,b_C^0$ but only during the first $\Delta_{\mathrm{ERI}}$ ms of the ERI; thereafter this build-up rate instantly recovers its initial value (so $b_C = b_C^0$) and then increases steadily, such that

$$b_C(t+\Delta t) = b_C(t) + a_{\mathrm{EX}}\,\Delta t \tag{6}$$

until the end of the ERI, where the term $a_{\mathrm{EX}}$ is the exogenous acceleration of the cue plan. In this way, the plan toward the cue, $r_C$, first halts for $\Delta_{\mathrm{ERI}}$ ms and then accelerates. If $r_C$ exceeds threshold during the ERI, a saccade toward the cue is triggered. Otherwise, the trial continues.

Epoch 3: after the ERI. During this last period, the plan toward the anti location first recovers its initial value (instantly, so $b_A = b_A^0$) and then accelerates, whereas the plan toward the cue decelerates. That is,

$$\begin{aligned} b_C(t+\Delta t) &= b_C(t) + d_{\mathrm{END}}\,\Delta t \\ b_A(t+\Delta t) &= b_A(t) + a_{\mathrm{END}}\,\Delta t \end{aligned} \tag{7}$$

where the endogenous deceleration $d_{\mathrm{END}}$ is negative and the endogenous acceleration $a_{\mathrm{END}}$ is positive. The process continues until one of the plans reaches the threshold.

Finally, the model also considers lapses, trials in which errors are made for reasons other than insufficient cue viewing time. Lapses occur with a probability $\lambda$, and are implemented as trials in which the endogenous acceleration and deceleration are equal to zero. In other words, a lapse corresponds to a trial in which the information about the correct target never reaches the circuit. During lapses, after the ERI (epoch 3), the motor plan toward the anti location continues building up at its initial rate, $b_A^0$, whereas the plan toward the cue continues advancing at whatever build-up rate it achieved at the end of the ERI.

In all, the model has 15 parameters that were adjusted to fit the pooled data set or the data from individual participants. Best-fitting values are listed in *Table 1* and *Supplementary file 1*. These were obtained by searching over a multidimensional parameter space, gradually reducing its volume, seeking to minimize the mean absolute error between the simulated and the experimental data. For each parameter vector tested, the error consisted of a sum of terms, each representing one target function to be fitted. These functions were the RT distributions for correct choices at individual gaps, the RT distributions for incorrect choices, also at individual gaps, and the tachometric curve. The search/minimization procedure was repeated multiple times with different initial conditions to ensure that solutions were found near the global optimum.

## Acknowledgements

We thank Ziad Hafed for comments and suggestions.

## Additional information

### Competing interests

Emilio Salinas: Reviewing editor, *eLife*. The other authors declare that no competing interests exist.

### Funding

| Funder | Grant reference number | Author |
| --- | --- | --- |
| National Eye Institute | R01EY025172 | Emilio Salinas Terrence R Stanford |
| National Institute of Neurological Disorders and Stroke | T32NS073553-01 | Christopher K Hauser |
| National Science Foundation | Graduate research fellowship | Christopher K Hauser |
| Tab Williams Family Endowment | | Emilio Salinas Terrence R Stanford |
| National Eye Institute | R01EY021228 | Emilio Salinas Terrence R Stanford |

The funders had no role in study design, data collection and interpretation, or the decision to submit the work for publication.

### Author contributions

Emilio Salinas, Conceptualization, Resources, Data curation, Software, Formal analysis, Supervision, Funding acquisition, Validation, Investigation, Visualization, Methodology, Writing—original draft, Project administration, Writing—review and editing; Benjamin R Steinberg, Software, Formal analysis, Investigation, Visualization; Lauren A Sussman, Sophia M Fry, Formal analysis, Investigation; Christopher K Hauser, Software, Formal analysis, Investigation, Methodology; Denise D Anderson, Supervision, Investigation, Project administration; Terrence R Stanford, Conceptualization, Resources, Data curation, Supervision, Funding acquisition, Validation, Methodology, Writing—original draft, Project administration, Writing—review and editing

### Author ORCIDs

Emilio Salinas (iD) https://orcid.org/0000-0001-7411-5693

### Ethics

Human subjects: All participants provided informed written consent before the experiment. All experimental procedures were conducted with the approval of the Institutional Review Board (IRB) of Wake Forest School of Medicine.

### Decision letter and Author response

Decision letter https://doi.org/10.7554/eLife.46359.027
Author response https://doi.org/10.7554/eLife.46359.028

## Additional files

### Supplementary files

• Source code 1. Matlab scripts and functions (.m) for running the accelerated race-to-threshold model as described in the article. It includes an example of the model output, a license agreement, and detailed instructions (README file).
DOI: https://doi.org/10.7554/eLife.46359.022

• Source data 1. Psychophysical data analyzed in this article.
DOI: https://doi.org/10.7554/eLife.46359.023

• Supplementary file 1. Parameters of the race-to-threshold model for individual participants.
DOI: https://doi.org/10.7554/eLife.46359.024

• Transparent reporting form
DOI: https://doi.org/10.7554/eLife.46359.025

### Data availability

The psychophysical data are provided in a supplementary data file (Source data 1). Matlab scripts for running the model are provided in a supplementary source code file (Source code 1).

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
