## [Decision Letter]

Thank you for submitting your article "Timescales of exogenous and endogenous attention revealed during urgent antisaccade performance" for consideration by *eLife*. Your article has been reviewed by three peer reviewers, including Daeyeol Lee as the Reviewing Editor and Reviewer #1, and the evaluation has been overseen by Timothy Behrens as the Senior Editor.

The reviewers have discussed the reviews with one another and the Reviewing Editor has drafted this decision to help you prepare a revised submission.

Summary:

The authors have developed an ingenious task that combined two well-studied previous paradigm, i.e., anti-saccade task and compelled saccade task. A tachometric curve shows that saccades generated prematurely tended to be directed incorrectly towards the cue, rapidly switching to the correct antisaccades. An interesting finding in this paper is that this switch occurs very rapidly, and the so-called attentional vortex lasts only about 40 ms. The authors have extended their previous race model, which accounts for the behavioral data very well, and also identify the individual variability lies mostly with how much the rate variables corresponding to the exogenous and endogenous plans accelerates immediately and some time after the cue onset, respectively.

Essential revisions:

1) The title implies that this is a study of attention and, indeed, the authors are using overt attention as their behavioral metric. However, some reviewers opined that the work primarily gives insight into possible underlying neural mechanisms driven by exogenous and endogenous processes. Whether these are synonymous with attention (whatever attention is) is debatable, as the section in the Discussion implies. Therefore it is suggested that the study should be framed in terms of processes and leave the word attention for tidbit in the Discussion.

The paper would be clearer (and more accurate) if the writing were more careful in its use of terminology, especially in the Results section. Although it is true that reaction time measurements have often been used as ways to measure "attention", this might not be appropriate anymore given the careful distinctions that have been drawn in the literature between effects on sensory processing versus motor preparation versus decision criteria, etc. These distinctions are especially relevant given the framing of this work, which builds on the idea that sensory processing and motor preparation proceed in parallel. To refer to effects on reaction time as due to aspects of "attention" or "perception" therefore seems inappropriate, since some of these effects might be attributable (and are, in fact, from the perspective taken in the authors' model) to motor preparation, saccade trigger thresholds, and possibly other non-sensory processes.

The phrase "attention vortex" is certainly memorable, but these effects may not all be due to changes in "attention" but also involve changes in saccade and fixation control. For example, it is relevant that the timing of the fixation offset is similar to the values that are also associated with the gap effect and express saccades – these are understood to be due to both the loss of visual drive for maintaining fixation as well as the addition of visual drive for the saccade. It seems likely that these types of motor preparatory factors are also involved here, and they deserve to be mentioned.

2) In various places in the manuscript, the authors describe behavior as resulting from a strong involuntary capture (e.g. subsection “Antisaccade performance varies with cue luminance”, first paragraph) or rates of activity buildup (e.g. subsection “Attentional capture as a perceptuo-motor phenomenon”, first paragraph). This might be questioned. The likely reason for the behavior seen is due to the urgency in the task. It is likely that rPTs that are less than 90 ms or that are in the vortex occur primarily because the gap is really long, not because one or another rate was high in the pre-ERI period. It would be important for the authors to not overly interpret their results. As they know, when the task is not compelled, these results tend not to be seen.

3) The modeling needs a stronger justification. The subdivision of time in the model into epochs with different properties seems ad hoc. The valuable thing about race models is that they provide an explanation for temporal aspects of behavior by defining the rates of hypothetical underlying processes. But this model includes explicit time-related variables (for example, defining the Exogenous Response Interval) to explain the temporal features of your data. It seems the interesting features of the data set are baked in. It's also disappointing that different sets of parameters are needed to explain the effects of the cue luminance manipulation.

Is it possible to define one rate process that would generate pro-saccades (perhaps shorter latency and lower rate), and a second rate process (perhaps longer latency and higher rate) that would generate the anti-saccades, and then allow them to interact so as to explain performance over time? The ERI would then emerge as a byproduct of the difference in latencies and rates between the two processes. Moreover, the rates could be defined as a function of the cue luminance, so that the scaling of the curves across cue luminance conditions might also emerge with one set of parameters. A model with this structure would be more parsimonious.

Along these same lines, it would be helpful to use the reaction time data from an independent set of pro-saccade trials to help constrain the rate parameters in the model. The parameters that capture the RT distribution for prosaccades might be expected to also explain the timing of the "dip" in the tachometric curve in the urgent anti-saccade task. It would be nice corroboration to see that this works on a per subject basis, but not if you switch pro-saccade rates between subject.

4) One of the three summary measurements for the tachometric curves is the "mean accuracy", and then several paragraphs are used to explain the seemingly counterintuitive relationship between mean accuracy and other aspects of performance. This aspect of the Results seemed poorly motivated, because I felt that "mean accuracy" was itself a bad measurement, because the underlying distribution tends to be bimodal (with modes at 0.5 and 1.0) so reporting the mean is a bad idea to begin with.

[Editors' note: further revisions were requested prior to acceptance, as described below.]

Thank you for resubmitting your work entitled "Voluntary and involuntary contributions to perceptually guided saccadic choices resolved with millisecond precision" for further consideration at *eLife*. Your revised article has been favorably evaluated by Timothy Behrens as the Senior Editor, a Reviewing Editor, and two reviewers.

The manuscript has been improved but one of the reviewers still has concerns that you might want to address before acceptance, as outlined below. Please note that there was some disagreement among the reviewers regarding how to best address these issues, but I hope that you would take them in account to finalize the manuscript.

1) Reviewers had additional discussion after they completed their individual reviews, regarding whether the use of the phrase "attentional vortex" was entirely appropriate. On the one hand, we all agree that the term "attention" might not be ideal, but we are also sympathetic to the reasons why you might prefer this term to an alternative, such as oculomotor capture, which might be technically more appropriate. The following is a comment from one of the reviewers on this issue:

The authors have been more careful in their use of loaded terms like attention and perception, although they still like to use the term "attention vortex". There are a few reasons why I still argue against using this term. First, as the authors say in their Reply: "the existing term 'oculomotor capture' describes exactly what happens in their task". What they have very nicely mapped out is the time course of oculomotor capture during the urgent antisaccade task. There is no need to introduce a new term. Second, the measurements here are oculomotor measurements, and the link to attention is only inferred. We do not know if there are also effects on visual perceptual judgments in addition to eye movements (for example, effects on visual discrimination). Third, the term attentional vortex implies that the capture is driven by attention, when really it is due to the interaction between saccade planning and changes in the state of attention (you could have called it "saccade vortex" with equal mixed justification). Fourth, isn't the attention field confusing already? You have the opportunity to put forward a clarifying description of how attention-related and saccade-related processes interact. Why ruin it? In short, I strongly recommend replacing "attention vortex" and "vortex" with "oculomotor capture", for the sake of properly and accurately acknowledging how these results fit into the existing literature.

2) Another issue about which the reviewers couldn't reach a completely consensus is whether and how the results related to the "mean accuracy" should be improved. While we all agree that you can probably perform additional analyses to clarify this part of the manuscript further, we did not agree on whether this would entirely necessary or not. The following is a comment from one of the reviewers:

The documentation of the tachometric curves (subsection “Antisaccade performance varies with cue luminance”, last paragraph) still seems to miss the mark for me, especially the section that goes into the problems with "mean accuracy". From the author's reply, I appreciate the point they are trying to make, but I think there may be other ways to achieve this goal.

There are actually two different points to be made. The first point is that the tachometric curves are not the same between subjects, and this presumably is due to differences in the temporal dynamics of the underlying sensory and saccade-related processes. This is an interesting point and backed up the second and third measurements drawn from the tachometric curves.

The second point is that mean accuracy can be misleading, or is at least incomplete, because it fails to register the separate effects that contribute to the dip in the tachometric curve. This is an important point, but I think you could make it much more directly. Here is my specific suggestion/request. When you refer to other studies using mean accuracy (subsection “Individual differences in perceptual and overall performance”, first paragraph), I presume you are referring to studies that used particular "gap" values, most often 0 milliseconds. With your own data, you could show how the distributions of correct and incorrect responses land on that subject's tachometric curve – for that one particular gap value – and what the "mean accuracy" would be for that subject if that was the only condition you had run. This would show more directly how the value of "mean accuracy" depends on the reaction time of the subject (as well as the gap value chosen), and how the tachometric curves captures these dynamics in ways that get obscured in the "mean accuracy" measurement. If this type of demonstration could explain the variation in previous studies using the anti-saccade task, that would be an especially useful addition.

---

## [Author Response]

Essential revisions:1) The title implies that this is a study of attention and, indeed, the authors are using overt attention as their behavioral metric. However, some reviewers opined that the work primarily gives insight into possible underlying neural mechanisms driven by exogenous and endogenous processes. Whether these are synonymous with attention (whatever attention is) is debatable, as the section in the Discussion implies. Therefore it is suggested that the study should be framed in terms of processes and leave the word attention for tidbit in the Discussion.The paper would be clearer (and more accurate) if the writing were more careful in its use of terminology, especially in the Results section. Although it is true that reaction time measurements have often been used as ways to measure "attention", this might not be appropriate anymore given the careful distinctions that have been drawn in the literature between effects on sensory processing versus motor preparation versus decision criteria, etc. These distinctions are especially relevant given the framing of this work, which builds on the idea that sensory processing and motor preparation proceed in parallel. To refer to effects on reaction time as due to aspects of "attention" or "perception" therefore seems inappropriate, since some of these effects might be attributable (and are, in fact, from the perspective taken in the authors' model) to motor preparation, saccade trigger thresholds, and possibly other non-sensory processes.The phrase "attention vortex" is certainly memorable, but these effects may not all be due to changes in "attention" but also involve changes in saccade and fixation control. For example, it is relevant that the timing of the fixation offset is similar to the values that are also associated with the gap effect and express saccades – these are understood to be due to both the loss of visual drive for maintaining fixation as well as the addition of visual drive for the saccade. It seems likely that these types of motor preparatory factors are also involved here, and they deserve to be mentioned.

We generally agree with this comment, and acknowledge that while our results are pertinent to various aspects of spatial attention, the word attention is heavily loaded and may refer to a variety of phenomena. Thus, we have changed the title and eliminated the reference to attention in it; as suggested, it now refers to voluntary and involuntary “processes”, which is more general. Similar, broader terminology (e.g., “endogenous control”, “endogenous responses”, etc.) was adopted at other points in the text as well. In addition, we revised the text to make the language more accurate, as suggested. In particular, the link between endogenous attention and the behavior we describe was certainly unclear. We have eliminated all references to endogenous attention, save to say that it may contribute in part to the endogenous process that directs the eyes away from the cue in our task.

That said, we think that there is generally much less ambiguity in the literature regarding the exogenous effects produced by salient, abrupt-onset stimuli. In fact, an earlier version of the manuscript was criticized precisely because we did not refer to the dip in the tachometric curve simply as attentional/oculomotor capture. Operationally, the existing term “oculomotor capture” describes exactly what happens in our task. By extension, the term “attentional capture” is also pertinent, for two reasons. First, because there is much evidence to indicate that oculomotor and attentional capture are indeed different expressions of the same underlying dynamics (not only psychophysical work by Theeuwes and others, but also neurophysiological studies; for instance, Dorris et al., 2007; Busse et al., 2008). And second, because our modeling results are entirely consistent with this idea: the exogenous effect is always the same (transient, biased acceleration/halting during the ERI), only sometimes it triggers a (captured) saccade immediately, and sometimes it does not — but in all likelihood, in the latter case the subthreshold increase in oculomotor activity corresponds to an increase in spatial attention. For these reasons, we believe that describing our findings in reference to attentional/oculomotor capture is not inappropriate. As for the closely related term “attentional vortex,” it is now unambiguously defined as the range of processing times in which saccades are highly likely to be captured. While captured saccades are not a new finding, their tightly constrained temporal window is.

Aside from the terminology, we entirely agree that captured saccades result from a combination of visually-evoked and motor preparatory activity that is facilitated by the early release of fixation. We now point this out along with the similarity between our captured saccades and express saccades, which demonstrate many mechanistic parallels (subsection “Attentional capture as a perceptuo-motor phenomenon”, first and last paragraphs).

2) In various places in the manuscript, the authors describe behavior as resulting from a strong involuntary capture (e.g. subsection “Antisaccade performance varies with cue luminance”, first paragraph) or rates of activity buildup (e.g. subsection “Attentional capture as a perceptuo-motor phenomenon”, first paragraph). This might be questioned. The likely reason for the behavior seen is due to the urgency in the task. It is likely that rPTs that are less than 90 ms or that are in the vortex occur primarily because the gap is really long, not because one or another rate was high in the pre-ERI period. It would be important for the authors to not overly interpret their results. As they know, when the task is not compelled, these results tend not to be seen.

What we meant was that both our findings and the results of previous experiments reporting oculomotor capture are likely caused by similar, low-level visual representations (acting on ongoing motor activity). We did not mean to imply that the two phenomena are identical (that, we don’t know). This is now stated more accurately (subsection “Antisaccade performance varies with cue luminance”, first paragraph).

As for the first paragraph of the subsection “Attentional capture as a perceptuo-motor phenomenon”, *both* the gap length and the intensity of the build-up process (i.e., urgency) determine the outcome of each trial. For instance, a captured saccade may result when the initial build-up rate is moderate and the gap is long, or when the initial build-up rate is high and the gap is short – but in either case the rPT will be around 111 ms. This is what the paragraph intends to point out, that the rPT marks the convergence of sensory (viewing duration) and motor (initial build-up) conditions that promote the capture. That said, the reviewer’s comment is definitely well taken, and appreciated. In that paragraph in particular the importance of the interplay between the two factors was not apparent. The paragraph has been rewritten to avoid oversimplification; it now reflects the more nuanced situation, in which both the gap and the build-up rate are important.

3) The modeling needs a stronger justification. The subdivision of time in the model into epochs with different properties seems ad hoc. The valuable thing about race models is that they provide an explanation for temporal aspects of behavior by defining the rates of hypothetical underlying processes. But this model includes explicit time-related variables (for example, defining the Exogenous Response Interval) to explain the temporal features of your data. It seems the interesting features of the data set are baked in. It's also disappointing that different sets of parameters are needed to explain the effects of the cue luminance manipulation.Is it possible to define one rate process that would generate pro-saccades (perhaps shorter latency and lower rate), and a second rate process (perhaps longer latency and higher rate) that would generate the anti-saccades, and then allow them to interact so as to explain performance over time? The ERI would then emerge as a byproduct of the difference in latencies and rates between the two processes. Moreover, the rates could be defined as a function of the cue luminance, so that the scaling of the curves across cue luminance conditions might also emerge with one set of parameters. A model with this structure would be more parsimonious.

The development of our model has been data driven, and as far as we can tell, its neurophysiological implications are accurate.

First, we emphasize that this model is 95% the same as that developed previously for an urgent color discrimination task (the current model contains the earlier one as a special case). The broad structure has been thoroughly validated against neurophysiological data recorded in FEF (Stanford et al., 2010; Salinas et al., 2010; Costello et al., 2013; Scerra et al., 2019). That work indicates that, initially, after the go signal is detected, the two alternative motor plans indeed advance with fixed, randomly sampled build-up rates, and that this independent build-up ends when perceptual information arrives at the motor circuit to accelerate the motor plan associated with the target and decelerate the plan associated with the distracter (Stanford et al., 2010; Salinas et al., 2010; Costello et al., 2013). Notably, in that earlier model the ERI existed already, except that it consisted of a brief, symmetric halting of *both* motor plans (this is now mentioned in the first paragraph of the subsection “The accelerated race-to-threshold model”). At the time, this feature was introduced into the model to account for a small but reliable dip in the distributions of processing times. More recently, we realized that a biased halting of the motor activity (more prolonged for the side opposite to the cue) during this pre-existing period, and as observed in so-called “saccadic inhibition” experiments, could partially explain the attentional vortex. The point is that the ERI is not an ad hoc assumption of the antisaccade model, but a manifestation of a normally subtle yet extraordinarily reliable effect of visual onsets (discussed at length by Salinas and Stanford, 2018). The temporal features of our data may seem odd because they are absent in other, non-urgent decision-making tasks that unfold over several hundreds of milliseconds — but such features are not “baked in,” they are simply the most consistent with our current and previous results, and with a large literature on the interaction between visual onsets and saccade planning (reviewed by Salinas and Stanford, 2018).

Second, although we did not fully understand the alternative modeling scenario outlined by the reviewer, we stress that the luminance results not only did not require additional assumptions or special tweaks to the model, but also are entirely consistent with the neurophysiological effects of luminance. The model certainly cannot account for the effect of luminance without some variation in parameter values. In the scenario outlined by the reviewer, “the rates could be defined as a function of the cue luminance.” But the build-up rates cannot depend on luminance before the cue information arrives at the model circuit. The rates do change with cue luminance (gradually, via acceleration and deceleration), but in order to do so the afferent delay inevitably associated with the cue onset must elapse; that latency defines the beginning of the ERI. Again, this afferent delay is a real constraint, not an arbitrary assumption.

The model was developed to fit the high-luminance data, and doing so required exogenous halting and acceleration during the ERI, as described (Figure 7). Afterward, once those mechanisms were in place, we found that the model also produced excellent fits to the medium- and low-luminance data. Moreover, the best-fit parameter values it found make perfect sense: according to the model, the main consequence of lowering the luminance of the cue is to increase the afferent delay associated with the detection of the cue by ∼50 ms. This is highly consistent with psychophysical data (Purushothaman et al., 1998; White et al., 2008) and neurophysiological reports (Bisley et al., 2004; van Rossum et al., 2008; Oram, 2010; Marino et al., 2015) showing that visual latencies everywhere, from retina to high-order visual areas to oculomotor structures, increase as luminance and/or contrast diminishes, with differences as large as 100 ms or more. The results for medium and low luminance confirm that the two mechanisms proposed initially are consistent with the temporal properties of visually sensitive neurons in oculomotor areas, which are thought to mediate exogenous attention. This is now mentioned in the fifth paragraph of the subsection “A comprehensive account of antisaccade behavior based on motor competition”.

And third, experiments are underway in our laboratory to directly probe the reality of the ERI, i.e., the discrete transition from uninformed build-up, to exogenous bias, to endogenous guidance of the saccadic choice. We are in the very early stages of this effort, but so far the results are encouraging. For example, preliminary data show that the exogenous modulations of activity during the ERI associated with stimulus detection can be manipulated independently of the later endogenous signal that informs the choice — but such manipulation requires high temporal precision, as expected from the short duration of the ERI.

Along these same lines, it would be helpful to use the reaction time data from an independent set of pro-saccade trials to help constrain the rate parameters in the model. The parameters that capture the RT distribution for prosaccades might be expected to also explain the timing of the "dip" in the tachometric curve in the urgent anti-saccade task. It would be nice corroboration to see that this works on a per subject basis, but not if you switch pro-saccade rates between subject.

The contrast between pro- and antisaccades would indeed be very informative. In fact, for an experiment with interleaved pro- and anti- trials, once the model is fitted to the antisaccade trials, all the prosaccade results become fully predictable. That is, having set all the parameter values using the anti data, the pro data turn into a parameter-free test of the model. This test is, in fact, particularly rich in our urgent task, because the predictions include not only a full tachometric curve (Author response image 1, red curve) but also specific shapes for the rPT distributions; for instance, the rPT histograms for correct pro (Author response image 1, red curve) and incorrect anti trials (Author response image 1, gray histogram) should overlap substantially up to the point where the pro curve reaches asymptotic performance. This, however, is a new experiment that goes beyond the scope of the present study.

**Author response image 1. respfig1:** Predictions for an experiment in which urgent prosaccades and antisaccades are interleaved. All results are from model simulations.

4) One of the three summary measurements for the tachometric curves is the "mean accuracy", and then several paragraphs are used to explain the seemingly counterintuitive relationship between mean accuracy and other aspects of performance. This aspect of the Results seemed poorly motivated, because I felt that "mean accuracy" was itself a bad measurement, because the underlying distribution tends to be bimodal (with modes at 0.5 and 1.0) so reporting the mean is a bad idea to begin with.

Thanks for noting the lack of clarity on this issue. In our data set, it may seem somewhat obvious that using the mean accuracy is not the best thing to do, but our point is precisely that this is not a good idea *in general*, regardless of whether one uses an urgent or a non-urgent version of the task. From our results, one can readily identify why this the case, and it boils down to two main reasons: first, the mean accuracy can be easily traded with speed (mean RT), and second, under non-urgent conditions the mean accuracy will largely reflect the rate of lapses, not the rate of captured saccades (which are the types of error that, implicitly or explicitly, are supposed to be characteristic of the antisaccade task). We have made various changes to the corresponding section (subsection “Individual differences in perceptual and overall performance”) to motivate the results better and state these points more clearly.

[Editors' note: further revisions were requested prior to acceptance, as described below.]The manuscript has been improved but one of the reviewers still has concerns that you might want to address before acceptance, as outlined below. Please note that there was some disagreement among the reviewers regarding how to best address these issues, but I hope that you would take them in account to finalize the manuscript.1) Reviewers had additional discussion after they completed their individual reviews, regarding whether the use of the phrase "attentional vortex" was entirely appropriate. On the one hand, we all agree that the term "attention" might not be ideal, but we are also sympathetic to the reasons why you might prefer this term to an alternative, such as oculomotor capture, which might be technically more appropriate. The following is a comment from one of the reviewers on this issue:The authors have been more careful in their use of loaded terms like attention and perception, although they still like to use the term "attention vortex". There are a few reasons why I still argue against using this term. First, as the authors say in their Reply: "the existing term 'oculomotor capture' describes exactly what happens in their task". What they have very nicely mapped out is the time course of oculomotor capture during the urgent antisaccade task. There is no need to introduce a new term. Second, the measurements here are oculomotor measurements, and the link to attention is only inferred. We do not know if there are also effects on visual perceptual judgments in addition to eye movements (for example, effects on visual discrimination). Third, the term attentional vortex implies that the capture is driven by attention, when really it is due to the interaction between saccade planning and changes in the state of attention (you could have called it "saccade vortex" with equal mixed justification). Fourth, isn't the attention field confusing already? You have the opportunity to put forward a clarifying description of how attention-related and saccade-related processes interact. Why ruin it? In short, I strongly recommend replacing "attention vortex" and "vortex" with "oculomotor capture", for the sake of properly and accurately acknowledging how these results fit into the existing literature.

We appreciate the reviewer’s point of view and the overall sentiment — that conservative interpretations are generally preferred. We think that our results do speak to the close relationship between attention-related and saccade-related processes, but we agree that using the term “attentional vortex” is perhaps not the best way to convey the specific implications of our results, particularly in the Results section. Since the most problematic part of the term is the qualifier, which brings in the baggage associated with attention, we have decided to omit the reference to attention and simply call it the vortex. At many points in the text we refer specifically to the part of the tachometric curve where captured saccades prevail, so giving it a name is a necessity. While “vortex” recalls the strong, involuntary nature of the capture, it is entirely neutral in regard to the attention versus oculomotor distinction.

In addition, we again reviewed all mentions to attention and attentional capture. A couple of them were either omitted or substituted with “oculomotor capture,” as suggested. Finally, we think that the exogenous increase in activity driven by the detection of the cue does correspond to the reflexive, covert allocation of attention – when that activity stays below threshold and does not trigger a saccade to the cue – but this interpretation is now more clearly articulated as such in the last part of the Discussion.

2) Another issue about which the reviewers couldn't reach a completely consensus is whether and how the results related to the "mean accuracy" should be improved. While we all agree that you can probably perform additional analyses to clarify this part of the manuscript further, we did not agree on whether this would entirely necessary or not. The following is a comment from one of the reviewers:The documentation of the tachometric curves (subsection “Antisaccade performance varies with cue luminance”, last paragraph) still seems to miss the mark for me, especially the section that goes into the problems with "mean accuracy". From the author's reply, I appreciate the point they are trying to make, but I think there may be other ways to achieve this goal.There are actually two different points to be made. The first point is that the tachometric curves are not the same between subjects, and this presumably is due to differences in the temporal dynamics of the underlying sensory and saccade-related processes. This is an interesting point and backed up the second and third measurements drawn from the tachometric curves.The second point is that mean accuracy can be misleading, or is at least incomplete, because it fails to register the separate effects that contribute to the dip in the tachometric curve. This is an important point, but I think you could make it much more directly. Here is my specific suggestion/request. When you refer to other studies using mean accuracy (subsection “Individual differences in perceptual and overall performance”, first paragraph), I presume you are referring to studies that used particular "gap" values, most often 0 milliseconds. With your own data, you could show how the distributions of correct and incorrect responses land on that subject's tachometric curve – for that one particular gap value – and what the "mean accuracy" would be for that subject if that was the only condition you had run. This would show more directly how the value of "mean accuracy" depends on the reaction time of the subject (as well as the gap value chosen), and how the tachometric curves captures these dynamics in ways that get obscured in the "mean accuracy" measurement. If this type of demonstration could explain the variation in previous studies using the anti-saccade task, that would be an especially useful addition.

The reviewer raises an interesting point about why exactly the mean accuracy fails to reflect the conflict between endogenous and exogenous influences that is evidenced by the tachometric curve. What we are asking in the subsection “Individual differences in perceptual and overall performance”, is indeed, whether using the mean accuracy to measure that conflict is a good idea. The reviewer notes an important nuance about our analysis: the results could depend on which gap values are used in the calculation. This was certainly worth looking into. However, it turns out that, regardless of which gaps we include in the analysis, the mean accuracy never correlates with the measures of perceptual performance derived from the tachometric curve.

In Figure 5, the mean accuracy and mean RT were computed using all the positive gap values, i.e., the same trials used to compute the tachometric curve. Calculated in this way, the mean accuracy is unrelated to perceptual accuracy (Figure 5A), and clearly demonstrates a speed-accuracy trade-off across individual participants (Figure 5B).

The same analysis was repeated based just on the zero-gap data, as suggested (Author response image 2). Again, there is no obvious relationship between observed accuracy and perceptual accuracy (Author response image 2), and because the former is generally very high at zero gap (mostly above 90% correct), the speed-accuracy trade-off is no longer visible (Author response image 2). The relationship between perceptual accuracy and mean RT is preserved (Author response image 2), but that just means that the RTs of zero-gap trials are highly consistent with those of nonzero-gap trials (i.e., the participants that respond quickly, do so for all gaps).

The zero-gap data are not particularly informative because, as can be seen in Figure 2—figure supplement 1G, H, they miss the vortex and predominantly reflect asymptotic performance (delay trials are even worse, because they *exclusively* cover the asymptotic range; Figure 2—figure supplement 1C, D). However, our initial explanation based on the speed-accuracy trade-off still stands: even when multiple gaps are used and the data cover the full rPT range (as in Figure 5), the mean accuracy is still not a reliable way to compare the perceptual abilities of two participants because each of them will generally sample their respective tachometric curves differently, according to their own urgency. Both factors are important, the gaps used and the unpredictable sampling bias due to urgency.

We thank the reviewer for raising this issue; it helped us articulate these results better. The last paragraph of the subsection “Individual differences in perceptual and overall performance”, was rewritten to explain the distinct roles of the two factors, gaps and urgency. The last paragraph of the subsection “Antisaccade performance in relation to cognitive conflict”, in the Discussion, elaborates on this point and was also revised with the reviewer’s comment in mind.

**Author response image 2. respfig2:** Same format as in Figure 5 of the main text, except that the mean observed accuracy and the mean RT were computed using zero-gap trials only. Perceptual accuracy values are the same as in Figure 5.